# Polyphasic Identification of *Distoseptispora* with Six New Species from Fresh Water

**DOI:** 10.3390/jof8101063

**Published:** 2022-10-10

**Authors:** Huang Zhang, Rong Zhu, Yun Qing, Hao Yang, Chunxue Li, Gennuo Wang, Di Zhang, Ping Ning

**Affiliations:** 1Shandong Provincial Key Laboratory of Water and Soil Conservation and Environmental Protection, College of Resources and Environment, Linyi University, West Side of North Section of Industrial Avenue, Linyi 276000, China; 2Faculty of Environmental Science and Engineering, Kunming University of Science & Technology, Kunming 650500, China; 3Faculty of Food Science and Engineering, Kunming University of Science & Technology, Kunming 650500, China; 4Key Laboratory for Humid Subtropical Eco-Geographical Processes of the Ministry of Education, Fujian Normal University, Fuzhou 350007, China

**Keywords:** eight new taxa, Distoseptisporaceae, morphology, phylogenetic analysis, submerged wood, taxonomy, sporidesmium-like taxa

## Abstract

Twelve new specimens of sporidesmium-like taxa were collected from freshwater habitats in China and Thailand. Phylogenetic analysis of nuc 28S rDNA (LSU), internal transcribed spacer (ITS), translation elongation factor 1-alpha (*TEF1*-α) and second-largest subunit of RNA polymerase II (*RPB2*) sequence data, combined with morphological data, revealed that they are *Distoseptispora* species. Among them, six new species, including *D**. aqualignicola*, *D**. aquamyces*, *D. crassispora*, *D. curvularia, D. nonrostrata* and *D**. pachyconidia*, are introduced. Two new combinations, *D. adscendens* and *D. leonensis*, are transferred from *Ellisembia*. A new habitat and geographical record are reported for *D. clematidis*, collected from a freshwater habitat in China. New *RPB2* sequence data for *D. dehongensis* are provided.

## 1. Introduction

Freshwater fungi have been reported in the Chytridiomycota, Zygomycota, Ascomycota and Basidiomycota, and their asexual morphs are the most widely distributed groups in fresh water [1]. In previous studies, sporidesmium-like hyphomycetes were commonly found on submerged wood in streams [2,3,4,5,6,7,8,9]. *Sporidesmium* was established by Link [10] in the early 1800s, with *S. atrum* as the type species. Ellis [11,12] retrieved and redescribed the genus as having unbranched conidiophores; integrated, terminal, monoblastic, determinate or percurrent conidiogenous cells; and acrogenous, solitary, transversely septate or distoseptate conidia. According to the broad circumscription, more than 400 *Sporidesmium* species were introduced, mostly based on morphology alone. To deal with a heterogeneous assemblage of species under *Sporidesmium*, Subramanian [13] reassessed the genus and introduced seven new genera, namely, *Acarocybellina*, *Ellisembia*, *Gangliophora*, *Hemicorynesporella*, *Penzigomyces*, *Repetophragma* and *Stanjehughesia*, to accommodate some species on the basis of conidial septation (euseptate vs. distoseptate) and conidiogenesis. Wu and Zhuang [14] placed *Penzigomyces* in *Sporidesmium* and *Imicles* in *Ellisembia*, thereby expanding the generic concepts of *Sporidesmium* (euseptate) and *Ellisembia* (distoseptate) to include fungi with no percurrent conidiophores or lageniform, doliiform or nodose percurrently extending conidiophores.

With increasing molecular data, multi-locus phylogenetic relationships between *Sporidesmium* and *Ellisembia* species have been investigated to gain a better understanding of their taxonomy, and it turned out that these species were distributed in different subclasses in Sordariomycetes, including Diaporthomycetidae, Hypocreomycetidae, Sordariomycetidae and Xylariomycetidae [5,7]. Therefore, some previously described species were revised and transferred to newly established genera, such as *Pseudosporidesmium*. The newly collected sporidesmium-like taxa were placed in new genera, such as *Pseudostanjehughesia* and *Distoseptispora*, and *Ellisembia* was synonymized with *Sporidesmium* [7].

*Distoseptispora* was introduced for sporidesmium-like species with macronematous, unbranched, olive-green, cylindrical conidiophores; monoblastic, integrated, determinate, terminal, cylindrical conidiogenous cells; and acrogenous, distoseptate, cylindrical, smooth, darker conidia with slightly paler, but not hyaline, rounded apices of indeterminate length. Su et al. [7] introduced two new species, *D. fluminicola* (type) and *D. aquatica*, found on submerged wood in China, and also included two taxa, *Ellisembia adscendens* and *E. Leonensis*, named *Distoseptispora adscendens* and *D. leonensis* in their phylogenetic tree, but did not validly introduce them as new combinations. Subsequently, the generic concept of *Distoseptispora* was emended to include percurrently elongating conidiogenous cells and euseptate, verrucose conidia with percurrent proliferation, based on the newly introduced species *D. guttulata* and *D. suoluoensis* [5]. In the past five years, more than 40 new species (excluding *D. submersa*) were successively reported in Thailand and China [3,5,7,8,15,16,17,18,19,20,21,22,23,24,25]. *Distoseptispora submersa*, found on submerged wood in China, was introduced by Luo et al. [17]. The authors stated that it was phylogenetically close to *D. tectonae* but had larger conidiophores and shorter conidia. However, Dong et al. [9] synonymized *D. submersa* with *D. tectonae* due to the small nucleotide differences discovered between these two species and identification as *D. tectonae* extending to larger conidiophores.

Recently, Yang et al. [24] used multi-locus analysis and reported, for the first time, a sexual morph in *Distoseptispora*, namely, *D. hyalina*, which was characterized by an immersed to semi-immersed, subglobose to ellipsoidal, dark-brown perithecial with a short neck; a relatively thick peridium; sparse, persistent, hyaline paraphyses; cylindrical asci with non-amyloid apical annuli; and hyaline, filiform, 0–3-septate ascospores with mucilaginous sheaths. To date, it is the only known sexual morph in *Distoseptispora*. The hosts of *Distoseptispora* species are diverse, including *Carex* sp., *Pandanus* sp. and *Tectona grandis*, as well as bamboo [3,18,19]. *Distoseptispora* is the only genus in Distoseptisporaceae, Distoseptisporales, Diaporthomycetidae and Sordariomycetes. 

In this study, 12 fresh specimens of sporidesmium-like taxa were collected from freshwater habitats in China and Thailand and were preliminarily identified as *Distoseptispora* species. In order to clarify the classification of these specimens, an updated phylogeny of Distoseptisporales and their relatives is provided based on a concatenated nuc 28S rRNA (LSU)–internal transcribed spacer (ITS)–translation elongation factor 1-alpha (*TEF1*-α)–second-largest subunit of RNA polymerase II (*RPB2*) dataset. Six new species and two new combinations are introduced in this study based on phylogenetic analyses combined with morphological data. The characters of species are discussed.

## 2. Materials and Methods

### 2.1. Sample Collection, Morphological Studies and Isolation

In this study, decaying wood samples were collected from various regions of Thailand (Chiang Rai Province) and China (Sichuan and Yunnan Provinces). Decaying wood samples were place in zip-lock plastic bags with sterile wet cotton. The locations, collectors, dates of collection and countries were recorded for all collections, and they were taken to the laboratory for detailed observation and the isolation of pure cultures.

Fungal structures were observed and examined using a Nikon SMZ–171 dissecting microscope. Micro-morphological characters were examined using a Nikon ECLIPSE Ni compound microscope and photographed with a Canon 600D digital camera fitted to the microscope. The Tarosoft (R) Image Frame Work program and the Oplenic program were used to measure the fungal structures. Adobe Photoshop CS6 extended v. 13.0 software (Adobe Systems, San Jose, CA, USA) was used to process images. Single-spore isolation was obtained following the method described in Chomnunti et al. [26]. Germinated spores were aseptically transferred to potato dextrose agar (PDA) and incubated at 20–25 °C for 2–4 weeks. Colony characteristics were recorded, including color and shape. Dry specimens were deposited in the herbarium of Mae Fah Luang University (MFLU), Chiang Rai, Thailand, and the Kunming Institute of Botany Academia Sinica (HKAS), Kunming, China. Living cultures were deposited in the Mae Fah Luang University Culture Collection (MFLUCC) and the Kunming Institute of Botany Culture Collection (KUMCC). Facesoffungi and Index Fungorum numbers were acquired according to Jayasiri et al. [27] and Index Fungorum (http://www.indexfungorum.org/names/names.asp, accessed on 15 March 2021), respectively.

### 2.2. DNA Extraction, PCR Amplification and Sequencing

Fresh fungal mycelia were scraped off with a needle and transferred into a 1.5 mL centrifuge tube. A Plant genomic DNA extraction kit (generic) (TreliefJM, Kunming, China) was used to extract DNA from fresh mycelia, according to the manufacturer’s instructions.

DNA amplification for LSU, ITS, *TEF1*-α and *RPB2* genes was performed by polymerase chain reaction (PCR). The primers and PCR thermal cycle programs are shown in Table 1. PCR products were confirmed on 1% agarose electrophoresis gels stained with ethidium bromide. PCR products were sent to TsingKe Biological Technology (Kunming, China) for purification and sequencing.

### 2.3. Phylogenetic Analyses

The raw sequences were initially checked with Finch TV v. 1.4.0, and each gene was subjected to a BlastN search in NCBI’s GenBank to assess the confidence level. Sequences of Distoseptisporaceae species, along with the newly generated sequences and related taxa, following Dong et al. [9] and Phukhamsakda et al. [25], were downloaded from GenBank. The gene sequences were aligned using the online multiple alignment program MAFFT v. 7 (http://mafft.cbrc.jp/alignment/server/index.html, accessed on 7 April 2022) [32] and manually improved, where necessary, in BioEdit v. 7.2 [33], then concatenated with Mesquite v. 3.11. 

Maximum likelihood (ML) and Bayesian inference (BI) methods were used for the phylogenetic analyses. Maximum likelihood analysis was performed with RAxML-HPC v. 8 on XSEDE in the CIPRES Science Gateway (https://www.phylo.org/portal2/home.action/, accessed on 8 April 2022) [34,35], with the following changes made to the default settings: maximum hours to run: 10; model for bootstrapping phase: GTRGAMMA; analysis type: rapid bootstrap analysis/search for best-scoring ML tree (-f a); bootstrapping type: rapid bootstrapping (-x); bootstrap iterations: 1000 (the maximum value allowed).

For BI analysis, the best model for each gene was selected using MrModelTest 2.3 [36], and GTR+I+G was selected as the best-fitting model for LSU, ITS *TEF1*-α and *RPB2*. BI analysis was conducted using MrBayes 3.2.6 for posterior probabilities (PPs) by Markov chain Monte Carlo sampling (BMCMC) [37,38]. Six simultaneous Markov chains were run for one million generations, and trees were sampled every 100 generations. The first 2500 trees, representing the burn-in phase of the analyses, were discarded and the remaining trees were used for the calculation of PPs in the majority rule consensus tree (the critical value for the topological convergence diagnostic was 0.01). The phylogenetic tree was visualized in FigTree v. 1.4.2 [39] and edited in Adobe Illustrator CS6 (Adobe Systems, USA). Newly generated sequences were submitted to GenBank. All of the sequences used in this study are listed in Table 2. 

## 3. Results

### 3.1. Phylogenetic Analyses 

The combined LSU, ITS, *TEF1*-α and *RPB2* dataset was analyzed for species of Distoseptisporales and their relatives, using *Myrmecridium banksiae* (CPC 19852) and *Myrmecridium*
*schulzeri* (CBS 100.54) as outgroup taxa. The dataset consists of 94 strains with an alignment length of 3743 total characters. The RAxML analysis resulted in a best-scoring likelihood tree selected with a final ML optimization likelihood value of −36376.671888. The matrix had 1984 distinct alignment patterns, with 35.27% undetermined characters or gaps. Estimated base frequencies were as follows: A = 0.241400, C = 0.263989, G = 0.281489, T = 0.213122; substitution rates AC = 1.230237, AG = 3.213547, AT = 1.297497, CG = 0.852333, CT = 6.671646, GT = 1.000000; gamma distribution shape parameter α = 0.254430. The phylogenetic trees generated from BI analyses were topologically similar to the one generated via the ML analyses, and the latter is shown in Figure 1.

The genus *Distoseptispora* consists of three well-supported clades: nine of our new isolates clustered in Clade 1, and another three clustered in Clade 3. Six of them were identified as new species which combined the morphological characteristics described below, viz., *D. aquamyces*, *D. aqualignicola*, *D. crassispora*, *D. curvularia*, *D. nonrostrata* and *D. pachyconidia*. MFLU 19–0544 and MFLU 17–1671 clustered with *D. dehongensis*, with strong support (100% MLBS/1.00 BIPP). KUNCC 21–10732, KUNCC 21–10727, MFLU 19–0543 and KUNCC 21–10728 clustered with *D. bambusae*, *D. clematidis*, *D. rayongensis* and *D. tectonae*, respectively, with high bootstrap support. 

### 3.2. Taxonomy in Distoseptispora

***Distoseptispora adscendens*** (Berk.) R. Zhu & H. Zhang, comb. nov. (Figure 2).

*Basionym*: *Sporidesmium adscendens* Berk., Ann. nat. Hist., Mag. Zool. Bot. Geol. 4: 292 (1840).

*Synonym*: *Clasterosporium adscendens* (Berk.) Sacc., Syll. fung. (Abellini) 4: 394 (1886).

*Ellisembia adscendens* (Berk.) Subram., Proc. Indian natn Sci. Acad., Part B. Biol. Sci. 58(4): 183 (1992).

*Sexual morph: Miyoshiella triseptata* (Shoemaker & G.P. White) Réblová, Mycotaxon 71: 33 (1999).

*Synonym*: *Lasiosphaeria triseptata* Shoemaker & G.P. White, Sydowia 38: 282 (1986).

*Index Fungorum number*: 559917; *Facesoffungi number*: 12574.

*Notes: Sporidesmium adscendens* was synonymized as *Ellisembia adscendens* on the basis of pseudoseptate conidia and conidiophores with irregular percurrent proliferation [13]. An ascomycetes, *Miyoshiella triseptata (≡Lasiosphaeria triseptata),* collected in Hungary, was found to be associated with *S. adscendens* in the host of *Platanus orientalis* and therefore reported as its sexual morph [40,41]. Shenoy et al. [42] reported a strain, HKUCC 10820, of *E. adscendens* with LSU and *RPB2* sequence data. The strain was later proved to phylogenetically cluster in the genus *Distoseptispora* by Su et al. [7] and was therefore named *Distoseptispora adscendens* in their phylogenetic tree but not validly introduced as a new combination. Many subsequent authors followed them in using the name *D. adscendens* [5,8,17,23].

The morphology of strain HKUCC 10820 was referred to Wu and Zhuang [[14], P107], who described it as having macronematous, mononematous, cylindrical, straight or slightly flexuous, smooth, 1–4-septate, brown to dark-brown conidiophores 20–50 μm long, 7–10 μm wide; integrated, terminal, cylindrical, smooth, brown to dark-brown conidiogenous cells with a size of 5–10 × 3–4.5 μm; and acrogenous, cylindrical to obclavate, rostrate, 16–62-distoseptate conidia 110–375 μm long, 14–20 μm wide at the broadest part, with rounded apexes 5–10 μm wide and subcylindrical to conical–truncate basal cells 5–7 μm wide at the base. This description is consistent with that of *S**. adscendens* given in Ellis [12], even with respect to the sizes of the conidiophores and conidia. The above characters fit the generic features of *Distoseptispora* well. Therefore, we formally place *E**. adscendens* in *Distoseptispora*, as *D. adscendens*.

***Distoseptispora aqualignicola*** C.X. Li & H. Zhang, sp. nov. (Figure 3).

*Index Fungorum number*: 559918; *Facesoffungi number*: 12575.

*Holotype*: HKAS 122184.

*Etymology*: “Aqua-*”* refers to the aquatic habitat in which the fungus was found, while “lignicola” refers to the fact that the fungus was found on a lignicolous substrate.

*Saprobic* on decaying wood submerged in freshwater. Sexual morph: undetermined. Asexual morph: hyphomycetous. *Colonies* on the substratum superficial, effuse, scattered, hairy, brown to dark-brown. *Mycelium* partly superficial, partly immersed, composed of branched, septate, smooth, hyaline to pale-brown hyphae. *Conidiophores* 90–190(–240) μm long ($\bar{x}$
*=* 162 µm, n = 15) and 5–8 μm wide ($\bar{x}$
*=* 6 µm, n = 15), macronematous, mononematous, unbranched $\bar{x}$, multi-septate, single or in groups of two or three, cylindrical, straight or slightly flexuous, smooth, brown, rounded at the apex. *Conidiogenous cells* 13–21 μm long ($\bar{x}$
*=* 18 µm, n = 10), 4–5.5 μm wide ($\bar{x}$ = 4.5 µm, n = 10), monoblastic, integrated, determinate, terminal, cylindrical, brown, smooth. *Conidia* 41–94(–104) μm long ($\bar{x}$ = 73 µm, n = 20) (rostrum included), 10.5–12.5 µm at the widest part ($\bar{x}$
*=* 11.5 µm, n = 20), 2–5 µm wide at the apex ($\bar{x}$
*=* 3.5 µm, n = 20), acrogenous, dry, obclavate, rostrate, straight or curved, 4–8-euseptate, mostly 6–7-euseptate, tapering towards the rounded apex, brown at the base, smooth, thin-walled, subhyaline to pale-brown at the apex.

*Culture characteristics*: Conidia germinating on PDA within 24 h. Germ tubes produced from the conidial apex. Colonies on PDA reaching 34 mm diameter after 22 days at 20–25 °C, circular, dry, dark-brown to black on surface and reverse, raised, with entire margin.

*Material examined*: China, Sichuan Province, Yibin City, Southern Sichuan Bamboo Sea, Qicai Lake, found on dead, submerged, decaying wood of unidentified plants, 16 June 2019, Chunxue Li, S1–10 (HKAS 122184, holotype), ex-type living culture KUNCC 21–10729.

*Notes: Distoseptispora aqualignicola, D**. aquamyces**,**D. lancangjiangensis,**D. meilingensis, D**. suoluoensis**, D**. yongxiuensis* and *D. verrucose* form a strongly supported clade (99% MLBS/1.00 BIPP) in Clade 2 of Distoseptispora (Figure 1). Morphologically, *D**. aqualignicola* possesses smooth-walled conidia, which are distinct from the verrucose conidia of *D. aquamyces,*
*D. suoluoensis* and *D. verrucosa* [5,24]. *Distoseptispora aqualignicola* differs from *D. meilingensis* in having euseptate, thin-walled conidia as compared with the distoseptate, thick-walled conidia of D. MEILINGENSIS [43]. *D**istoseptispora lancangjiangensis* and *D**. yongxiuensis* resemble *D**. aqualignicola* with respect to their smooth-walled conidia but they are phylogenetically distinct. 

***Distoseptispora aquamyces*** R. Zhu & H. Zhang, sp. nov. (Figure 4).

*Index Fungorum number*: 559919; *Facesoffungi number*: 12576.

*Holotype*: HKAS 122186.

*Etymology*: “Aqua-*”* refers to the aquatic habitat in which the fungus was found.

*Saprobic* on decaying wood submerged in freshwater. **Sexual morph:** undetermined. **Asexual morph:** hyphomycetous. *Colonies* on the substratum: superficial, effuse, scattered, hairy, brown to dark-brown. *Mycelium* partly superficial, partly immersed, composed of branched, septate, smooth, hyaline to pale-brown hyphae. *Conidiophores* (78–)91–198 μm long ($\bar{x}$
*=* 127 µm, n = 10), 4–7 μm wide ($\bar{x}$
*=* 5 µm, n = 10), macronematous, mononematous, unbranched, multi-septate, single or in groups of two or three, cylindrical, straight or slightly flexuous, smooth, pale-brown. *Conidiogenous cells* 10–20 μm long ($\bar{x}$
*=* 16 µm, n = 10), 4.5–5.5 μm wide ($\bar{x}$
*=* 5 µm, n = 10), monoblastic, integrated, determinate, terminal, cylindrical, pale-brown, smooth. *Conidia* 30–95 μm long ($\bar{x}$
*=* 73 µm, n = 20) (rostrum included), 7–12 µm at the widest part ($\bar{x}$
*=* 8 µm, n = 20), 2–5 µm wide at the apex ($\bar{x}$
*=* 3 µm, n = 20), acrogenous, dry, obclavate to obpyriform, mostly rostrate, straight or curved, 4–10-euseptate, mostly 6–9-euseptate, tapering towards the rounded apex, verrucose, thin-walled, pale-brown to brown at the base, subhyaline to pale-brown at the apex, verrucose.

*Culture characteristics:* Conidia germinating on PDA within 24 h. Germ tubes produced at end of conidia. Colonies on PDA reaching 40 mm diameter after 45 days at 20–25 °C, circular, flat, surface rough, grey from above, dark-brown from below, edge entire.

*Material examined*: China, Sichuan Province, Yibin City, Southern Sichuan Bamboo Sea, Qicai Lake, found on dead, submerged, decaying wood of unidentified plants, 3 November 2020, Yun Qing, SN–18 (HKAS 122186, holotype), ex-type living culture KUNCC 21–10731.

Notes: *Distoseptispora aquamyces* differs from *D. suoluoensis* in having pale-brown to brown, shorter conidia (30–95 μm vs. 80–125 μm), sometimes with percurrent proliferation, while *D. aquamyces* has yellowish-brown or dark-olivaceous conidia without proliferation. In our phylogenetic tree, *Distoseptispora aquamyces* is close to *D. suoluoensis*, with low support. Two species in *Distoseptispora* possess verrucose conidia, i.e., *D. suoluoensis* and *D. verrucose*. *Distoseptispora verrucose* has percurrently proliferating conidiogenous cells and olivaceous conidia which differ from the non-proliferating conidiogenous cells and pale-brown to brown conidia of *D. aquamyces*. More importantly, *D. aquamyces* is distinct from *D. suoluoensis* and *D. verrucose* in our phylogenetic tree. Therefore, *D. aquamyces* is introduced as a new species.

***Distoseptispora bambusae*** Y.R. Sun, I.D. Goonasekara, Yong. Wang bis & K.D. Hyde, Biodiversity Data Journal 8(e53678): 6 (2020) (Figure 5).

*Saprobic* on decaying wood submerged in freshwater. Sexual morph: undetermined. Asexual morph: hyphomycetous. *Colonies* on the substratum superficial, effuse, hairy, black. *Mycelium* mostly immersed, composed of branched, septate, smooth, pale-brown hyphae. *Conidiophores* (39–)45–90 μm long ($\bar{x}$ = 64 µm, n = 10), 3.5–5 μm wide ($\bar{x}$ = 4.5 µm, n = 10), macronematous, mononematous, septate, unbranched, single or in groups of two or three, cylindrical, straight or slightly flexuous, smooth, pale-brown, robust at the base. *Conidiogenous cells* 12–20 μm long ($\bar{x}$ = 15 µm, n = 10), 4–5 μm wide ($\bar{x}$ = 4 µm, n = 10), monoblastic or polyblastic, integrated, determinate, terminal, cylindrical, pale-brown, smooth. *Conidia* (29–)43–94(–105) μm long ($\bar{x}$ = 71 μm, n = 25) (rostrum included), 6–8.5 μm at the widest part ($\bar{x}$ = 7 μm, n = 25), 2–3 µm wide at the apex ($\bar{x}$ = 2.5 µm, n = 25), acrogenous, solitary, obclavate, rostrate, straight or curved, (4–)7–13-euseptate, rounded at the apex, truncate at the base, tapering towards apex, smooth, thin-walled, pale-brown or brown.

*Culture characteristics*: Conidia germinating on PDA within 24 h. Germ tubes produced from the two sides of the conidia. Colonies on PDA reaching 40 mm diameter after 40 days at 20–25 °C, circular, flat, rough surface, grey from above, dark-brown from below, edge entire. 

*Material examined*: China, Sichuan Province, Yinbin City, Southern Sichuan Bamboo Sea, Qicai Lake, found on dead, submerged, decaying wood of unidentified plants, 13 November 2020, Yun Qing, SN–34 (HKAS 122187), living culture KUNCC 21–10732. 

*Notes*: *Distoseptispora bambusae* was initially collected from culms of bamboo by Sun et al. [22] in Guizhou, China, and Chiangrai, Thailand. The morphological characters and size of our new collection are consistent with the holotype, except for the wider range of conidial length ((29–)43–94(–105) μm vs. 45–74 μm). Phylogenetically, the new collection clusters with *D. bambusae*, with strong support (100% MLBS/1.00 BIPP; Figure 1). Our isolate was collected from dead, submerged, decaying wood, which constitutes a new habitat record.

***Distoseptispora clematidis*** Phukhams., M.V. de Bult & K.D. Hyde, Fungal Diversity 102(1): 168 (2020) (Figure 6).

*Saprobic* on decaying wood submerged in freshwater. Sexual morph: undetermined. Asexual morph: hyphomycetous. *Colonies* on the substratum superficial, effuse, scattered, hairy, dark-brown. *Mycelium* mostly immersed, composed of branched, septate, smooth, dark-brown hyphae. *Conidiophores* (8–)15–25 μm long ($\bar{x}$ = 20 µm, n = 10), 5–8 μm wide ($\bar{x}$ = 7 µm, n = 10), macronematous, mononematous, solitary, unbranched, single or in groups of two or three, 2–5-septate, cylindrical, straight or slightly flexuous, smooth, dark-brown to brown, robust at the base. *Conidiogenous cells* 4–7 μm long ($\bar{x}$ = 5 µm, n = 10), 4–6 μm wide ($\bar{x}$ = 5 µm, n = 10), monoblastic, integrated, determinate, terminal, cylindrical, dark-brown, smooth. *Conidia* (90–)126–245(–303) µm long (rostrum included) ($\bar{x}$ = 185 µm, n = 30), 14–20 µm at the widest part ($\bar{x}$ = 16 µm, n = 30), 7–10 µm wide at the apex ($\bar{x}$ = 8.5 µm, n = 30), acrogenous, dry, obclavate, elongated, straight or curved, (17–)20–41(–49)-distoseptate, truncate at the base, rounded at the apex, smooth, thick-walled, brown with a green tinge, sometimes with percurrent proliferation and forming another conidium.

*Culture characteristics*: Conidia germinating on PDA within 24 h. Germ tubes produced from the conidial base. Colonies on PDA reaching 34 mm diameter after 3 weeks at 20–25 °C, brown with sparse mycelium from above, dark-brown from below, rough surface, dry, flat, entire at edge.

*Material examined*: China, Yunnan Province, Xishuangbanna, Man Feilong Reservoir, found on dead, submerged, decaying wood of unidentified plants, 7 November 2020, Rong Zhu, N76 (HKAS 122182), living culture KUNCC 21–10727.

*Notes: Distoseptispora clematidis* was introduced by Phukhamsakda et al. [21] and was found on a dried stem of *Clematis sikkimensis* in Thailand. The new isolate KUNCC 21–10727 clusters with the type strain of *D. clematidis*, with strong support (98% MLBS/1.00 BIPP; Figure 1). Morphologically, our isolate has the same characters as the holotype of *D. clematidis*, including conidial size and conidial septa number, except for slightly shorter conidiophores ((8–)15–25 μm vs. 22–40 μm)) [21]. The conidia of the new isolate sometimes have percurrent proliferation and form another conidium from the conidial apex, which was not reported in the original description of *D. clematidis*, though it is obvious in picture f of Figure 109 in Phukhamsakda et al.’s [21]. Comparisons of sequence data for KUNCC 21–10727 and the ex-type strain showed nine (1%, including one gap), nine (1.6%, no gaps) and five (0.7%, including one gap) nucleotide differences in the LSU, ITS and *RPB2* regions, respectively. Although the nucleotide differences in the ITS regions were more than 1.5%, which is high enough to establish a new species, according to Jeewon and Hyde [44], there were no significant morphological differences between our isolate and the holotype. Therefore, we propose to identify the new isolate as *D. clematidis* until more strains have been examined. The new isolate was collected from submerged, decaying wood in China, which constitutes a new habitat and geographical record.

***Distoseptispora crassispora*** R. Zhu & H. Zhang, sp. nov. (Figure 7).

*Index Fungorum number*: 559920; *Facesoffungi number*: 12577.

*Holotype:* HKAS 122181.

*Etymology*: “Crass” means thick and refers to the thick-walled conidia.

*Saprobic* on decaying wood submerged in freshwater. Sexual morph: undetermined. Asexual morph: hyphomycetous. *Colonies* on the substratum superficial, effuse, scattered, hairy, dark-brown. *Mycelium* mostly immersed, composed of branched, septate, smooth, dark-brown hyphae. *Conidiophores* 14–27 μm long, 6–10 μm wide, macronematous, mononematous, solitary, unbranched, septate, cylindrical, straight or flexuous, smooth, brown to dark-brown, robust at the base. *Conidiogenous cells* 2.5–7 μm long ($\bar{x}$ = 4.5 μm, n = 10), 5.5–8 μm wide ($\bar{x}$ = 6 μm, n = 10), monoblastic, integrated, determinate, terminal, cylindrical, dark-brown, smooth. *Conidia* 95–197(–214) µm long (rostrum included) ($\bar{x}$ = 141 μm, n = 30), 13–24 μm at the widest part ($\bar{x}$ = 17 μm, n = 30), 6–12 μm wide at the apex ($\bar{x}$ = 9 μm, n = 30), acrogenous, dry, obclavate, rostrate, mostly curved, 15–36(–41)-distoseptate, truncate at the base, rounded at the apex, smooth, thick-walled, brown with a green tinge.

*Culture characteristics*: Conidia germinating on PDA within 24 h. Germ tubes produced from the two sides of the conidia. Colonies on PDA reaching 20 mm diameter after 20 days at 20–25 °C, circular, raised, velvety, aerial, medium–sparse, dark-brown from both above and below.

*Material examined*: China, Yunnan Province, Xishuangbanna, Man Feilong Reservoir, found on dead, submerged, decaying wood of unidentified plants, 7 November 2020, Rong Zhu, N66 (HKAS 122181, holotype), ex-type living culture KUNCC 12–10726.

*Notes*: Multi-gene phylogenetic analyses showed that *D. crassispora* is a distinct species in *Distoseptispora* and clusters with *D. chinense* and *D. tectonigena* (97% MLBS/1.00 BIPP; Figure 1). *Distoseptispora crassispora* is morphologically similar to the latter two species in having straight to slightly curved, septate conidiophores and obclavate, distoseptate, rostrate, straight or slightly curved conidia. However, *D. crassispora* possesses shorter conidiophores (up to 27 μm vs. up to 110 μm) and wider conidia (13–24 μm vs. 10–13 μm) than *D. tectonigena*. Additionally, the conidia of *D. tectonigena* are sometimes percurrently proliferating 5–10 times at the apex, which was not observed in *D. crassispora* [15]. *Distoseptispora crassispora* can be distinguished from *D. chinense* on the basis of molecular data. Comparisons of nucleotides between *D. crassispora* and *D. chinense* revealed 12 (2.3%, including four gaps) and 23 (2.6%) nucleotide differences in the ITS region and *TEF1*-α, respectively, which follows the generally accepted norm, according to which nucleotide differences of more than 1.5% in the ITS region are likely to indicate a new species [44]. We therefore introduce *D. crassispora* as a new species in *Distoseptispora*.

***Distoseptispora curvularia*** R. Zhu & H. Zhang, sp. nov. (Figure 8).

*Index Fungorum number*: 559921; *Facesoffungi number*: 12578.

*Holotype*: HKAS 122180.

*Etymology*: In reference to the mostly curved conidia.

*Saprobic* on decaying wood submerged in freshwater. Sexual morph: undetermined. Asexual morph: hyphomycetous. *Colonies* on the substratum superficial, effuse, scattered, hairy, dark-brown. *Mycelium* mostly immersed, composed of branched, septate, smooth, dark-brown hyphae. *Conidiophores* 11–28 μm long ($\bar{x}$ = 18 µm, n = 10), 5–9 μm wide ($\bar{x}$ = 7 µm, n = 10), macronematous, mononematous, unbranched, septate, cylindrical, straight or flexuous, smooth, brown to dark-brown, robust at the base. *Conidiogenous cells* 5–6 μm long, 5–5.5 μm wide, monoblastic, integrated, determinate, terminal, cylindrical, dark-brown, smooth. *Conidia* (60–)100–200(–314) µm long (rostrum included) ($\bar{x}$ = 155 μm, n = 30), 12–19 μm at the widest part ($\bar{x}$ = 15 μm, n = 30), 7–9.5 μm wide at the apex ($\bar{x}$ = 9 μm, n = 30), acrogenous, dry, obclavate, rarely oblong, rostrate, mostly curved, (9–)16–48(–59)-distoseptate, truncate at the base, rounded at the apex, smooth, thick-walled, brown with a green tinge.

*Culture characteristics*: Conidia germinating on PDA within 24 h. Germ tubes produced from the conidial base. Colonies on PDA reaching 30 mm in diameter after 20 days at 20–25 °C, circular, raised, convex or dome-shaped with dark-brown papillate surface, brown at the margins, dark-brown in reverse. 

*Material examined*: China, Yunnan Province, Xishuangbanna, Man Feilong Reservoir, found on dead, submerged, decaying wood of unidentified plants, 7 November 2020, Rong Zhu, N53 (HKAS 122180, holotype), ex-type living culture KUNCC 21–10725.

*Notes*: The new species has a close phylogenetic affinity to *D. clematidis* (98% MLBS/1.00 BIPP; Figure 1). Morphologically, *D. curvularia* is similar to *D. clematidis* in having acrogenous, obclavate, rostrate conidia and being brown with a green tinge. However, *D. curvularia* differs from the holotype of *D. clematidis* (MFLU 17–1501) in having longer conidia (up to 314 μm vs. up to 210 μm) and more conidial septa (up to 59 vs. up to 35) [15]. Additionally, our new collection of *D. clematidis* (KUNCC 21–10725) possesses conidia sometimes with percurrent proliferation and the formation of another conidium from conidial apices. A comparison of sequence data for these two species showed a difference of 15 (2.81%, no gaps) noticeable nucleotides in ITS gene regions. Therefore, we introduce *D. curvularia* as a new species.

***Distoseptispora dehongensis*** W. Dong, H. Zhang & K.D. Hyde, Fungal Diversity 96, 145 (2019) (Figure 9 and Figure 10).

Specimen MFLU 19–0544: *Conidiophores* 70–110 μm long, 3–4.5 μm wide, 5–7-septate, pale-brown to mid-brown. *Conidiogenous cells* (12–)18–23 μm long, 3–4.5 μm wide, monoblastic or polyblastic, pale-brown. *Conidia* 20–57 μm long ($\bar{x}$ = 31 µm, n = 30), 6–15 μm at the widest part ($\bar{x}$ = 10 µm, n = 30), 5.5–7 μm wide at the apex ($\bar{x}$ = 6 μm, n = 20), acrogenous or intercalary, solitary or catenate, obpyriform to obclavate, broad cylindrical or irregular, non-rostrate, straight or curved, 4–7-distoseptate, mostly 6-distoseptate, truncate at the base, rounded at the apex, smooth, thick-walled, pale-brown, sometimes with percurrent proliferation and forming another conidium from the conidial apex.

Specimen MFLU 17–1671: *Conidiophores* 47–128 μm long, 3–5.5 μm wide, 4–7-septate, pale-olivaceous. *Conidiogenous cells* 18–25 μm long, 4–5 μm wide, monoblastic or polyblastic, pale-olivaceous. *Conidia* 17–80 μm long ($\bar{x}$ = 51 µm, n = 30), 8–13 μm at the widest part ($\bar{x}$ = 13 μm, n = 30), 4–6 μm wide at the apex ($\bar{x}$ = 5 μm, n = 30), acrogenous or intercalary, solitary, obpyriform, non-rostrate, straight or curved, 4–10-distoseptate, mostly 6–8-distoseptate, truncate at the base, smooth, thick-walled, olivaceous, sometimes with percurrent proliferation and forming another conidium from the conidial apex. 

*Material examined*: Thailand, Mukdahan Province, small river of Nong Bo Na Kae, found on dead, submerged, decaying wood of unidentified plants, 13 December 2018, Hao Yang, t34 (MFLU 19–0544), ex-type living culture MFLUCC 19–0335; Thailand, Long Khot subdistrict, found on submerged wood in a stream, 1 September 2017, Gennuo Wang, 4.7 (MFLU 17–1671), living culture MFLUCC 17–2326.

*Notes*: Our new isolates, MFLUCC 19–0335 (from MFLU 19–0544) and MFLUCC 17–2326 (from MFLU 17–1671), cluster with ex-type strain KUMCC 18–0090 of *D. dehongensis*, with strong support (100% MLBS/1.00 BIPP; Figure 1). Morphologically, the two new isolates fit well with the characters of *D. dehongensis* in having monoblastic or polyblastic conidiogenous cells and obpyriform to obclavate, straight or curved conidia with fewer than ten distosepta [45], except that the conidia of MFLU 19–0544 and MFLU 17–1671 show some percurrent proliferation, which is not observed in the holotype [45]. Additionally, the conidia of MFLU 19–0544 are pale-brown, while those of the holotype and MFLU 17–1671 are olivaceous. Comparisons of sequence data between the three strains showed no differences in LSU genes and three to six nucleotide differences in ITS genes. The holotype was collected from Yunan Provice, China [45], while our isolates were collected from Thailand, their locations constituting new geographic records. We provide new *RPB2* sequence data for *D. dehongensis* in this study.

***Distoseptispora leonensis*** (M.B. Ellis) R. Zhu & H. Zhang, comb. nov. (Figure 11).

*Basionym*: *Sporidesmium leonense* M.B. Ellis, Mycol. Pap. 70: 28 (1958).

*Synonym*: *Ellisembia leonensis* (M.B. Ellis) McKenzie, Mycotaxon 56: 13 (1995).

*Index Fungorum number*: 559922; *Facesoffungi number*: 12579.

*Notes*: The case of *Distoseptispora leonensis* is the same as that of *D. adscendens*, that is, it is the only strain of *D. leonensis* (HKUCC 10822) reported by Shenoy et al. [42], with LSU and *RPB2* sequence data, which has been proven to phylogenetically cluster in the genus *Distoseptispora* [7] and has been named as *D. adscendens* in the phylogenetic tree, though it has not been validly introduced as a new combination. The morphology of strain HKUCC 10822 was also referred to Wu and Zhuang [14] (P137), who described it as having macronematous, mononematous, cylindrical, straight or slightly flexuous, smooth, 5–7-septate, mid-brown to brown conidiophores 110–130 μm long, 6–8 μm wide; integrated, terminal, cylindrical, smooth, pale-brown to brown conidiogenous cells 5–15 × 4.5–5 μm in size; and acrogenous, obclavate, fusiform or ellipsoidal, rostrate, 8–10-distoseptate conidia, 50–75 μm long, 15–18 μm wide at the broadest part, with conical–truncate basal cells 3–4.5 μm wide at the base. This description is consistent with the description and illustration of the holotype of *S. leonensis* given in Ellis [11], even regarding the sizes of conidiophores and conidia, though the conidiophore and conidial proliferations are points of difference. The above characters fit the generic features of *Distoseptispora* well.

***Distoseptispora******nonrostrata*** Y. Qing & H. Zhang, sp. nov. (Figure 12).

*Index Fungorum number*: 559923; *Facesoffungi number*: 12580.

*Holotype*: HKAS 122185.

*Etymology*: In reference to the mostly non-rostrate conidia.

*Saprobic* on decaying wood submerged in freshwater. Sexual morph: undetermined. Asexual morph: hyphomycetous. *Colonies* on the substratum superficial, effuse, hairy or velvety, dark-brown. *Mycelium* mostly immersed, consisting of branched, septate, smooth, subhyaline to pale-brown hyphae. *Conidiophores* 105–160 μm long ($\bar{x}$ = 129 µm, n = 10), 4.5–7 μm wide ($\bar{x}$ = 5.5 µm, n = 10), macronematous, mononematous, solitary, unbranched, 5–10-septate, cylindrical, straight or slightly flexuous, smooth, brown, becoming pale-brown towards apex. *Conidiogenous cells* 15–20 μm long ($\bar{x}$ = 18 µm, n = 10), 5–5.5 μm wide ($\bar{x}$ = 5 µm, n = 10), monoblastic, integrated, determinate, terminal, cylindrical, pale-brown, smooth. *Conidia* 22–51 μm long ($\bar{x}$ = 36 μm, n = 20), 8–14 μm at the widest part ($\bar{x}$ = 12 μm, n = 20), 5.5–10 μm wide at the apex ($\bar{x}$ = 8 μm, n = 20), acrogenous, solitary, oblong, obclavate or narrowly obpyriform, mostly non-rostrate, rarely rostrate, straight or curved, 4–10-distoseptate, truncate at the base, smooth, thick-walled, pale-olivaceous or pale-brown.

*Culture characteristics*: Conidia germinating on PDA within 24 h. Germ tubes produced from conidial bases. Colonies on PDA reaching 30–35 mm diameter after 5 weeks at 20–25 °C, circular, raised, fluffy, dense, convex or dome-shaped, with dark-brown papillate surface, brown in the center, lighter on the outside, dark-brown at the margins, consistently dark-brown in reverse. 

*Material examined*: China, Sichuan Province, Yibin City, Southern Sichuan Bamboo Sea, Qicai Lake, found on dead, submerged, decaying bamboo, 13 November 2020, Yun Qing, SN–16 (HKAS 122185, holotype), ex-type living culture KUNCC 21–10730.

*Notes*: In our phylogenetic analyses, *Distoseptispora nonrostrata* cluster with *D. effusa*. Morphologically, *D. nonrostrata* possesses oblong, obclavate or narrowly obpyriform, mostly non-rostrate, pale-olivaceous or pale-brown conidia, while *D. effusa* is clearly different in having obclavate, rostrate, olivaceous-brown to dark-brown, longer conidia (35.5–113 μm vs. 22–51 μm) [24]. Comparisons of sequence data between these two species showed differences of 27 (4.9%, four gaps) and 34 (3.8%, no gaps) noticeable nucleotides in ITS and *TEF* gene regions, respectively. Therefore, we introduce *D. nonrostrata* as a new species.

***Distoseptispora pachyconidia*** R. Zhu & H. Zhang, sp. nov. (Figure 13).

*Index Fungorum number*: 559924; *Facesoffungi number*: 12581.

*Holotype*: HKAS 122179.

*Etymology*: “Pachy-” means thick in Latin, referring to the thick-walled conidia.

*Saprobic* on decaying wood submerged in freshwater. Sexual morph: undetermined. Asexual morph: hyphomycetous. *Colonies* on the substratum superficial, effuse, hairy or velvety, dark-brown or black. *Mycelium* mostly immersed, consisting of branched, septate, smooth, hyaline to pale-brown hyphae. *Conidiophores* 11–27 μm long ($\bar{x}$ = 19 µm, n = 10), 4–9 μm wide ($\bar{x}$ = 6 µm, n = 10), macronematous, mononematous, solitary, unbranched, 2–4-septate, cylindrical, straight or flexuous, smooth, pale-brown to mid-brown, slightly tapering distally, truncate at the apex. *Conidiogenous cells* 5–9 μm long ($\bar{x}$ = 7 µm, n = 10), 4–6.5 μm wide ($\bar{x}$ = 5 µm, n = 10), monoblastic, integrated, determinate, terminal, cylindrical, pale-brown, smooth. *Conidia* 42–136 μm long ($\bar{x}$ = 84 μm, n = 30), 14–22 μm at the widest part ($\bar{x}$ = 17 μm, n = 30), 8–14 μm wide at the apex ($\bar{x}$ = 11 μm, n = 30), acrogenous, solitary, obclavate, lanceolate, rostrate or not, straight or curved, 8–21-distoseptate, truncate at the base, tapering towards the rounded apex, smooth, thick-walled, pale-brown with a green tinge.

*Culture characteristics*: Conidia germinating on PDA within 24 h. Germ tubes produced from both ends. Colonies on PDA reaching 35–38 mm diameter after 5 weeks at 20–25 °C, dark-brown at the margins, greenish-glaucous at the center, olivaceous-brown in reverse, with smooth margin.

*Material examined*: China, Yunnan Province, Xishuangbanna, Na Dameng Reservoir, found on dead, submerged, decaying wood of unidentified plants, 7 November 2020, Rong Zhu, N37 (HKAS 122179, holotype), ex-type living culture KUNCC 21–10724.

*Notes*: *Distoseptispora pachyconidia* clusters as an independent branch in *Distoseptispora* based on a concatenated LSU–ITS–*TEF1*-α–*RPB2* phylogeny. It is phylogenetically close to *D. crassispora*, *D. chinense* and *D. tectonigena*. Compared to *D. tectonigena*, *D. pachyconidia* has shorter conidiophores (11–27 μm vs. up to 110 μm) and shorter conidia (42–136 μm vs. 148–225(–360) μm), as well as fewer conidial septa (8–21 vs. 20–46) [3]. In addition, the conidia of *D. tectonigena* were sometimes percurrently proliferating 5–10 times at the apex, which was not observed in *D. pachyconidia*. The conidia in *D. pachyconidia* are light-colored and shorter than those in *D. crassispora* and *D. chinense* (42–136 μm vs. 95–197 μm vs. 81–283) [46]. More importantly, *D. pachyconidia* is phylogenetically distinct from *D. crassispora*, *D. chinense* and *D. tectonigena.*

***Distoseptispora rayongensis*** J. Yang & K.D. Hyde, Mycosphere 11(1): 579 (2020) (Figure 14).

*Saprobic* on decaying wood submerged in freshwater. Sexual morph: undetermined. Asexual morph: hyphomycetous. *Colonies* on the substratum superficial, effuse, hairy or velvety, black. *Mycelium* mostly immersed, consisting of branched, septate, smooth, brown hyphae. *Conidiophores* 62–76 μm long ($\bar{x}$ = 69 µm, n = 10), 4–6 μm wide ($\bar{x}$ = 5.5 µm, n = 10), macronematous, mononematous, solitary, unbranched, 2–4-septate, cylindrical, straight or slightly flexuous, smooth, brown, rounded at apex. *Conidiogenous cells* 17–23 μm long, 5–6 μm wide, monoblastic, integrated, determinate, terminal, cylindrical, brown, smooth. *Conidia* 90–180(–261) μm long (rostrum included) ($\bar{x}$ = 134 µm, n = 20), 10–15 µm at the widest part ($\bar{x}$ = 13 µm, n = 20), 2–5 µm wide at the apex ($\bar{x}$ = 4 µm, n = 20), acrogenous, obclavate or obspathulate, rostrate, straight or slight curved, 12–17-distoseptate, truncate at the base, rounded at the apex, guttulate, smooth, thick-walled, pale-brown or pale-olivaceous, becoming paler or hyaline towards the apex.

*Culture characteristics*: Conidia germinating on PDA within 24 h. Germ tubes produced from the conidial apices. Colonies on PDA fast-growing, reaching 16 mm diameter after 15 days at 25 °C, circular, aerial mycelium dense, brown, reverse dark-brown, with entire, white margin.

*Material examined*: Thailand, Mukdahan Province, small river of Nong Bo Na Kae, found on dead, submerged, decaying wood of unidentified plants, 13 December 2018, Hao Yang, t33 (MFLU 19–0543), living culture MFLUCC 19–0334.

*Notes*: *Distoseptispora rayongensis* was introduced by Hyde et al. [47] and collected from a freshwater habitat in Rayong Province, Thailand. In our phylogenetic analysis, our new isolate was grouped with *D. rayongensis*, with strong support (99% MLBS/1.00 BIPP; Figure 1). Morphologically, our isolate was the same as *D. rayongensi*, except that it had shorter conidiophores (62–76 μm vs. 75–125 µm). Additionally, in the holotype, conidia showed percurrent proliferation and conidia forming from the conidial apices, which was not observed in our isolate. Comparisons of nucleotides between the holotype and our isolate showed no differences in ITS and *TEF1*-α regions and 1 and 35 (3.5%, including 21 gaps) nucleotide differences in LSU and *RPB2* regions, respectively.

***Distoseptispora tectonae*** Doilom & K.D. Hyde, Fungal Diversity 80: 222 (2016) (Figure 15).

*Saprobic* on decaying wood submerged in freshwater. Sexual morph: undetermined. Asexual morph: hyphomycetous. *Colonies* on the substratum superficial, effuse, scattered, hairy, black. *Mycelium* partly superficial, partly immersed, composed of branched, septate, smooth, pale-brown hyphae. *Conidiophores* 9–35 μm long, 5–7 μm wide, macronematous, mononematous, unbranched, 1–3-septate, cylindrical, straight or slightly flexuous, smooth, brown to dark-brown, robust at the base. *Conidiogenous cells* 5–10 μm long, 4.5–5.5 μm wide, holoblastic, monoblastic, integrated, terminal, cylindrical, brown, smooth. *Conidia* 46–192 μm long ($\bar{x}$ = 111 µm, n = 30) (rostrum included), 12–18 µm at the widest part ($\bar{x}$ = 15 µm, n = 30), 8–12 µm wide at the apex ($\bar{x}$ = 10 µm, n = 30), acrogenous, solitary, cylindric–obclavate, elongate, rostrate, straight or curved, 9–39-distoseptate, rounded at the apex, truncate at the base, smooth, thick-walled, olivaceous-green when young, dark-brown at maturity, with a wedge-shaped base, 3.5–6 μm wide.

*Culture characteristics*: Conidia germinating on PDA within 24 h. Germ tubes produced from the conidial apices. Colonies on PDA reaching 35 mm diameter after 36 days at 20–25 °C, circular, dry, dark-olivaceous-green to black on surface and reverse, raised, with entire margin.

*Material examined*: China, Sichuan Province, Yibin City, Southern Sichuan Bamboo Sea, Qicai Lake, found on dead, submerged, decaying wood of unidentified plants, 16 June 2019, Chunxue Li, S1–4 (HKAS 122183), living culture KUNCC 21–10728.

*Notes*: *Distoseptispora tectonae* was initially collected from a dead twig of *Tectona grandis* [3]. Subsequently, an isolate which was collected from submerged water and identified as *D. submerse* was synonymized as *D. tectonae* [9]. More recently, two new isolates collected from saprobic and submerged water in Thailand were reported [9,22]. Our new isolate clustered with the above strains, with strong support (100% MLBS/1.00 BIPP; Figure 1). It has a wider range of conidial lengths than the holotype (46–192 μm vs. (90–)130–140(–170) μm) [3]. The conidia of our isolate and MFLU 15–2693 are olivaceous, while those of the holotype (MFLU 15–3417) and MFLU 20–0262 are brown.

## 4. Discussion

To date, a total of 54 *Distoseptispora* species have been reported. They have been found to cluster into three well-supported clades, which is in agreement with all previous studies on *Distoseptispora* [5,8,9,23,24,25]. The morphologies of *Distoseptispora* species are similar, and taxonomic identifications mainly rely on phylogeny. To reappraising generic and specific delimitations, we summarized the characters of all *Distoseptispora* species in Figure 1, including the length of conidiophores; proliferation and conidiogenesis (monoblastic or polyblastic) in conidiogenous cells; the type (distoseptate or euseptate) and number of septa, shape, length, color, proliferation, rostrateness and wall thickness of conidia; as well as habitats. With respect to the characters listed in Figure 1, we can summarize that:

(1) Proliferation and conidiogenesis (monoblastic or polyblastic) in conidiogenous cells

Proliferation of conidiogenous cells has been reported in all clades: *D. amniculi*, *D. guttulata*, *D. effusa* and *D. rostrata* in Clade 1; *D. atroviridis* and *D. fusiformis* in Clade 2; and *D. verrucosa* and *D. yunnanensis* in Clade 3. Although no proliferation has been reported in *D. palmarum* and *D. yunnanensis*, we can see obvious proliferation in the illustrations of Hyde et al. [45] (Figure 104d) and Li et al. [23] (Figure 3b,h). All species possess monoblastic conidiogenous cells, except for *D. palmarum*, which possesses polyblastic ones, and *D. dehongensis* in Clade 1, *D. saprophytica* in Clade 2, and *D. bambusae* and *D. meilingensis* in Clade 3, which possess mono- or polyblastic ones.

(2) Type of conidial septa—distoseptate or euseptate

Regarding the definitions of “distoseptate” and “euseptate”, in *Dictionary of the Fungi* (9th edition), “euseptate (of conidial septation)” is defined as “having cells separated by multilayered walls of similar structure to lateral walls”, while “distoseptate (of conidial septation)” is defined as “having the individual cells each surrounded by a sac-like wall distinct from the outer wall”. Given these definitions and actual observations in *Distoseptispora* species, we can conclude that the euseptate condition appears as a single line or thicker region in the middle of septa, while the distoseptate condition is characterized by sac-like septa, usually narrower in the middle of septa or exhibiting cracks, as in *D. clematidis*, *D. crassispora* and *D. nonrostrata* (indicated by the arrows in Figure 6f, Figure 7d and Figure 12g in this paper).

According to Figure 1, most species in Clades 1 and 2 have distoseptate conidia, just like the generic type *D. fluminicola*. The species with euseptate conidia are all grouped in Clade 3, except for *D. guttulata* and *D. martinii*. *Distoseptispora martinii* was first described as *Acrodictys martinii*, then was transferred to *Junewangia* and *Rhexoacrodictys* [48,49,50]. Recently, it was proved to be a new combination in *Distoseptispora* on the basis of phylogenetic analysis [16]. Morphologically, it has ellipsoid, oblate to subglobose, muriform conidia instead of obclavate to obpyriform, transversely distoseptate conidia. *Distoseptispora uttulate* is another exception in *Distoseptispora*, as it has 11–14-euseptate conidia. *Distoseptispora lignicola* was described as euseptate, but it can be seen from the illustrations that it is euseptate only at the rostra. At the main part of the conidia, it should be distoseptate, due to the obvious narrowing in the middle of the septa [8] (Figure 5g,h). *Distoseptispora rayongensis* has the same characteristics, such that its septa should be recognized as distoseptate ([47], Figure 99k; this study, Figure 10g). *Distoseptispora meilingensis* is the only species in Clade 3 that has distoseptate conidia.

(3) Type of conidial wall—thick or thin, smooth or verrucose

All species in Clades 1 and 2 have thick-walled conidia, except for *D. guttulata* and *D. martini*, which have thin-walled conidia. The species with euseptate and thin-walled conidia are all grouped in Clade 3, except for *D. meilingensis*, which has thick-walled conidia. *Distoseptispora bambusae* and *D. euseptata* were described as thick-walled, with wall thicknesses less than 1 µm—thicker than those in *D. Suoluoensis* and *D. Lancangjiangensis* but not as thick as the ones in Clades 1 and 2 (ca. 2–4 µm). Thus, rather than thick-walled, it is more proper to call them medium–thick-walled. All *Distoseptispora* species have smooth conidial walls, except for *D. aquamyces*, *D. suoluoensis* and *D. verrucose*, which are verrucose-walled.

(4) Conidial shapes and color

Conidial shapes in Clades 1 and 2 are mostly obclavate, sometimes obpyriform, oblong, subcylindrical/cylindrical, rarely fusiform or ellipsoid/oblate/subglobose, while those in Clade 3 are obclavate. *D. euseptata* also has obpyriform conidia. Conidial color in most species is brown with a green tinge, and a few have a single color. For example, in Clade 1, conidia are brown in *D. tectonigena* and *D. longispora*, and olivaceous green in *D. multiseptata*. Conidial color may vary within a species. For example, in *D. tectonae*, conidia are brown in the holotype (MFLU 15–3417) and specimen MFLU 20–0262 and olivaceous in specimen MFLU 15–2693. Another example is *D. dehongensis*: conidial color is pale brown in specimen MFLU 19–0544 and olivaceous in the holotype and specimen MFLU 17–1671.

(5) The length of conidiophores and conidia, and numbers of septa

The lengths of conidiophores in the three clades are irregular, including 0 (maximum < 100 µm), 1 (minimum < 100 µm and maximum > 100 µm) and 2 (minimum > 100 µm). The conidial lengths of species in Clade 1 are relatively long, mostly longer than 100 µm, even up to 700 µm. Therefore, they have more conidial septa—many more than 20 and even up to 80. The conidial lengths of species in Clade 3 are relatively short, all mostly shorter than 100 µm. Conidial length may vary within a species, as in *D. multiseptata*. The conidial lengths of collection MFLU 15–1144 (mostly 300–600 µm long, up to 700 µm long) are significantly longer than those of the holotype (up to 380 µm long) [5]. Yang et al. [5] suspected that the length of conidia may depend on the length of incubation.

(6) Conidial rostrateness

Conidia are described as rostrate when they have a beak-like extension or appearance, according to *Dictionary of the Fungi* (9th edition), where “beak” means a pointed part on the tip (https://dictionary.cambridge.org, accessed on 8 May 2022). Given this definition, we re-checked all species from the published illustrations and listed those with rostrate conidia in Figure 1. Species with rostrate conidia that were not reported as such in their original descriptions are labeled in red. From Figure 1, we can see that most species possess rostrate conidia. Species with only non-rostrate conidia are *D. atroviridis*, *D. dehongensis*, *D. fusiformis*, *D. hydei*, *D. martinii* and *D. saprophytica*.

(7) Conidial proliferation

Conidial proliferation has been described in Clade 1 (*D. amniculi*, *D. bangkokensis*, *D. tectonigena*, *D. rayongensis*, *D. yunjushanensis*) and Clade 3 (*D. suoluoensis*). *Distoseptispora saprophytica* was thought to produce catenate conidia occasionally [9]. However, it is more proper to use the term conidial proliferation, because the upper conidia were obviously immature [9] (Figure 14g). Although no conidial proliferation was reported in *D. xishuangbannaensis* and *D. thailandica*, we can see obvious proliferation in their illustrations ([18], Figure 65c,d and Figure 64c). Conidial proliferation is not a species-delimiting factor because an individual species may exhibit variation in this character, as in *D. clematidis*, *D. dehongensis* and *D. multiseptata*. These species were not reported as being characterized by conidial proliferation with respect to their holotype, though proliferation was reported in the new collections ([18], Figure 3; this study, Figure 3 and Figure 7).

(8) Habitats

Most *Distoseptispora* species were reported from freshwater habitats, except for *D. adscendens*, *D. hydei*, *D. martinii*, *D. tectonigena*, *D. thailandica* and *D. xishuangbannaensis* in Clade 1, and *D. caricis*, *D. leonensis* and *D. palmarum* in Clade 2.

To sum up, no significant morphological differences were found among Clades 1, 2 and 3. Therefore, there is not enough evidence to segregate Clades 2 and 3 from Clade 1, in which the type species *D. fluminicola* is included, although Clades 1, 2 and 3 all received strong support (100% MLBS/1.00 BIPP; Figure 1). Conidial differentiation on the basis of euseptate and distoseptate does not receive molecular support, as such differences are found in both Clades 1 and 3. This is the same as the situation for euseptate/distoseptate conidia in *Sporidesmium*. Subramanian [13] segregated the species with distoseptate conidia from *Sporidesmium* and introduced the genus *Ellisembia* to accommodate them. However, Su et al. [7] proved that the euseptate/distoseptate difference between *Sporidesmium* and *Ellisembia* does not have molecular support and proposed *Ellisembia* as a synonym of *Sporidesmium*. Yang et al. [24] also inferred that euseptate/distoseptate differences have no taxonomic significance for generic delimitation of sporidesmium-like taxa, though they are informative at the species level.

## Figures and Tables

**Figure 1 jof-08-01063-f001:**
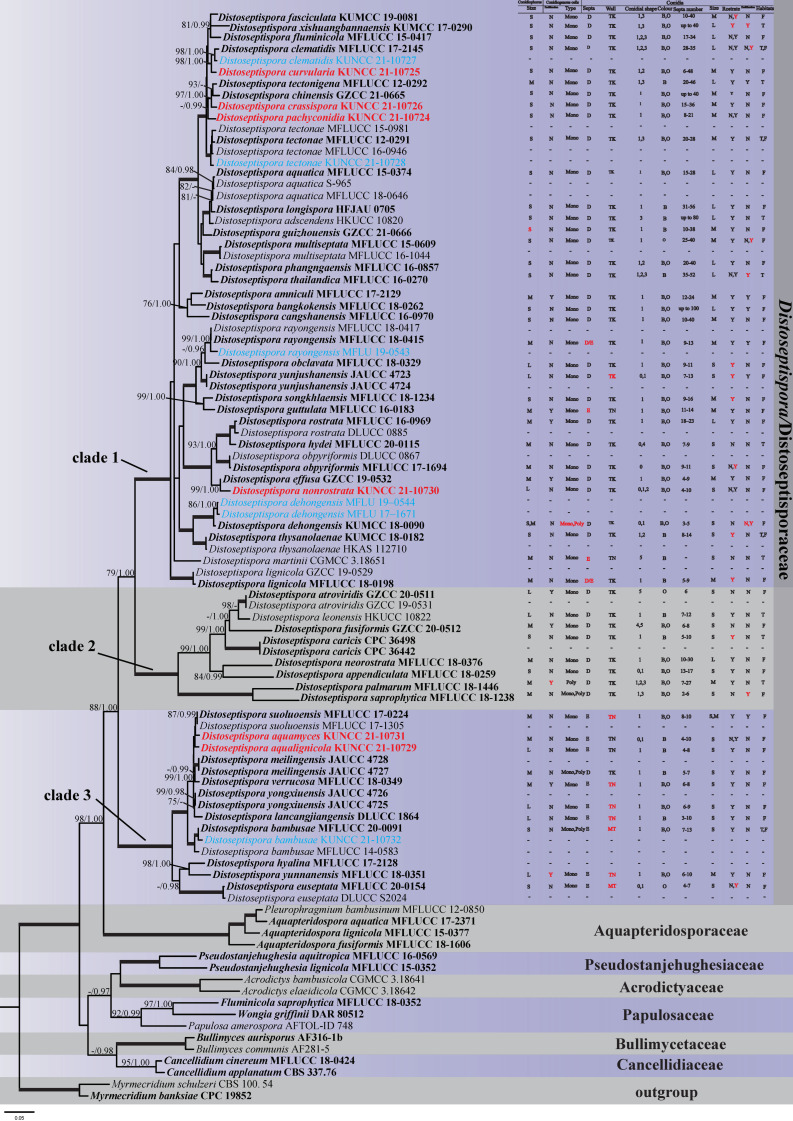
The best-scoring maximum likelihood (RAxML) tree for the family Distoseptisporaceae, based on the LSU, ITS, *TEF1*-α and *RPB2* sequence dataset. The tree is rooted with *Myrmecridium schulzeri* (CBS 100.54) and Myrmecridium banksiae (CPC 19852). Maximum likelihood support values greater than 70% and Bayesian posterior probabilities greater than 0.95 are shown near the nodes. New strains identified in this study are shown in blue; new species are shown in red. Ex-type strains are shown in bold. The characters of all *Distoseptispora* species are summarized, including conidiophores (size: S (maximum length < 100 µm), L (minimum length > 100 µm), M (minimum length < 100 µm and maximum length > 100 µm)); conidiogenous cells (proliferation: Y (yes), N (no); type: mono (monophyletic), poly (polyphyletic)); and conidia (septa: D (distospetate), E (euseptate); wall: TK (thick), TN (thin), MT (medium thick); conidial shape: 0 (obpyriform), 1 (obclavate), 2 (oblong), 3 (subcylindrical/cylindrical), 4 (fusiform), 5 (ellipsoid, oblate or subglobose); color: B (brown), O (olivaceous or green); septa number; size: S, M and L (the same as for conidiophore size); rostrateness and proliferation: Y (yes), N (no); habitats: T (terrestrial), F (freshwater)).

**Figure 2 jof-08-01063-f002:**
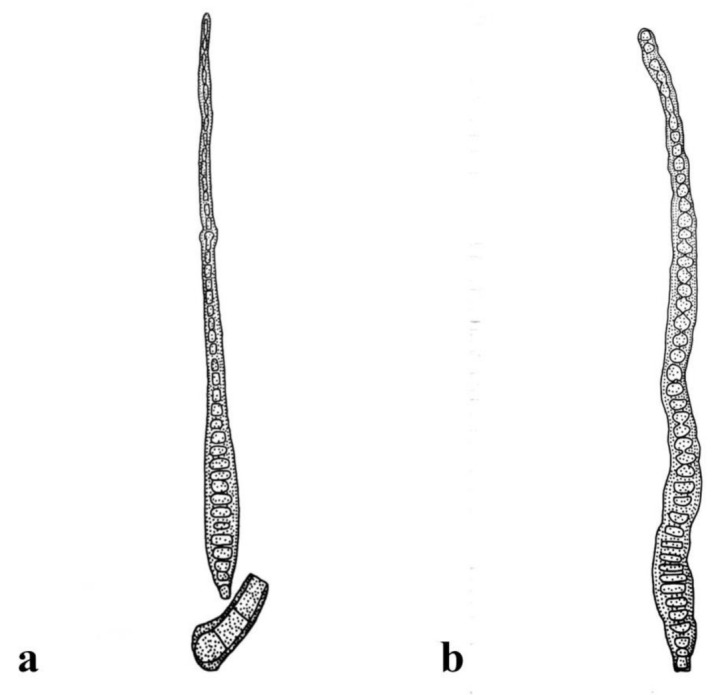
*Distoseptispora adscendens*. (redraw according to Wu and Zhuang [14]). (**a**) Conidiophores with conidia. (**b**) Conidia.

**Figure 3 jof-08-01063-f003:**
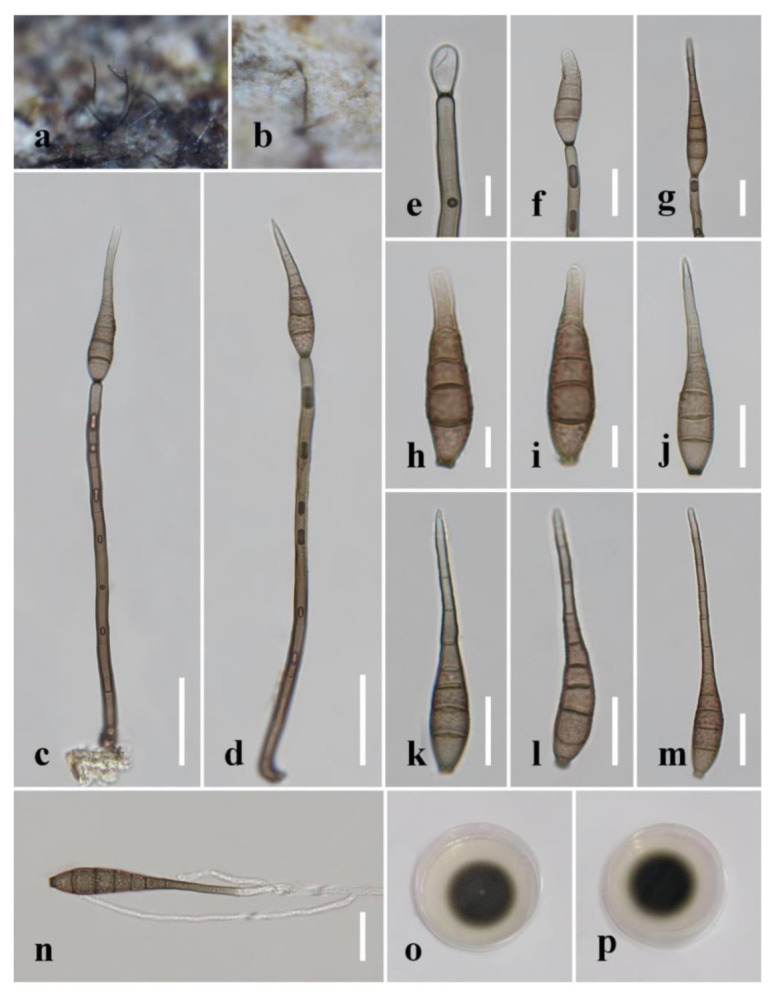
*Distoseptispora aqualignicola* (HKAS 122184, holotype). (**a**,**b**) Colonies on natural substrate. (**c**,**d**) Conidiophores with conidia. (**e**–**g**) Conidiogenous cells bearing conidia. (**h**−**m**) Conidia. (**n**) Germinated conidium. (**o**) Colony on PDA (from front). (**p**) Colony on PDA (from reverse). Scale bars: (**c**,**d**) 30 μm; (**e**,**h**–**i**) 10 μm; (**f**,**g**,**j**–**n**) 20 μm.

**Figure 4 jof-08-01063-f004:**
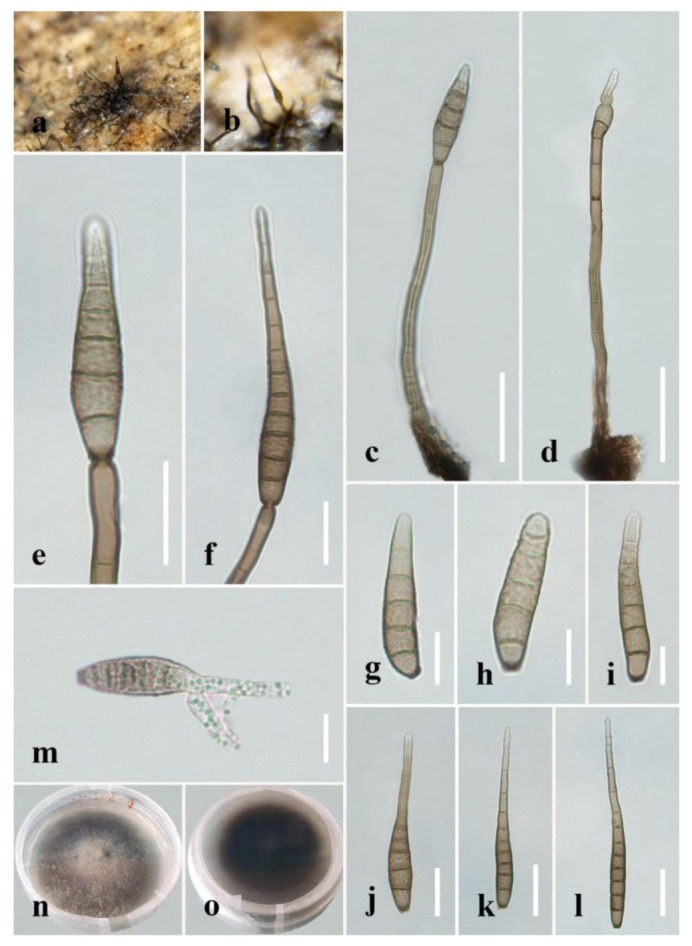
*Distoseptispora aquamyces* (HKAS 122186, holotype). (**a**,**b**) Colonies on natural substrate. (**c**,**d**) Conidiophores with conidia. (**e**,**f**) Conidiogenous cells bearing conidia. (**g**−**l**) Conidia. (**m**) Germinated conidium. (**n**) Colony on PDA (from front). (**o**) Colony on PDA (from reverse). Scale bars: (**c**,**d**) 40 μm; (**e**,**f**,**j**–**l**) 20 μm; (**g**–**i**,**m**) 10 μm.

**Figure 5 jof-08-01063-f005:**
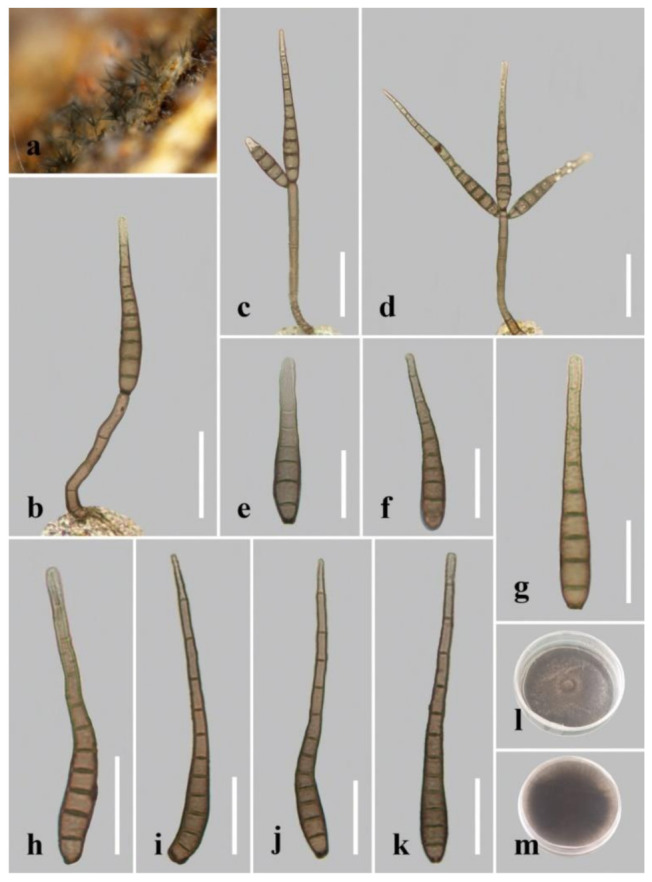
*Distoseptispora bambusae*. (**a**) Colonies on natural substrate. (**b**–**d**) Conidiophores with conidia. (**e**–**k**) Conidia. (**l**) Colony on PDA (from front). (**m**) Colony on PDA (from reverse). Scale bars: (**b**–**d**) 30 μm; (**e**–**k**) 20 μm.

**Figure 6 jof-08-01063-f006:**
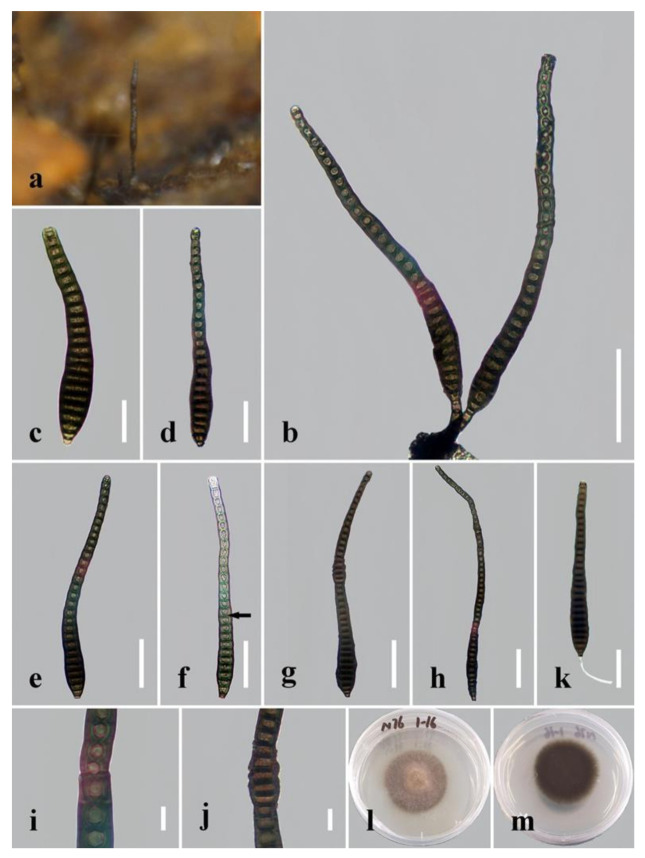
*Distoseptispora clematidis*. (**a**) Colonies on natural substrate. (**b**) Conidiophores with conidia. (**c**–**h**) Conidia. The arrow in **f** points to a gap in the middle of the septum, which indicates the distosepta. (**i**,**j**) Portions of conidia with percurrent proliferation in the middle of conidia. (**k**) Germinated conidium. (**l**) Colony on PDA (from front). (**m**) Colony on PDA (from reverse). Scale bars: (**b**,**g**–**h**) 50 μm; (**c**,**d**) 10 μm; (**i**,**j**) 5 μm.

**Figure 7 jof-08-01063-f007:**
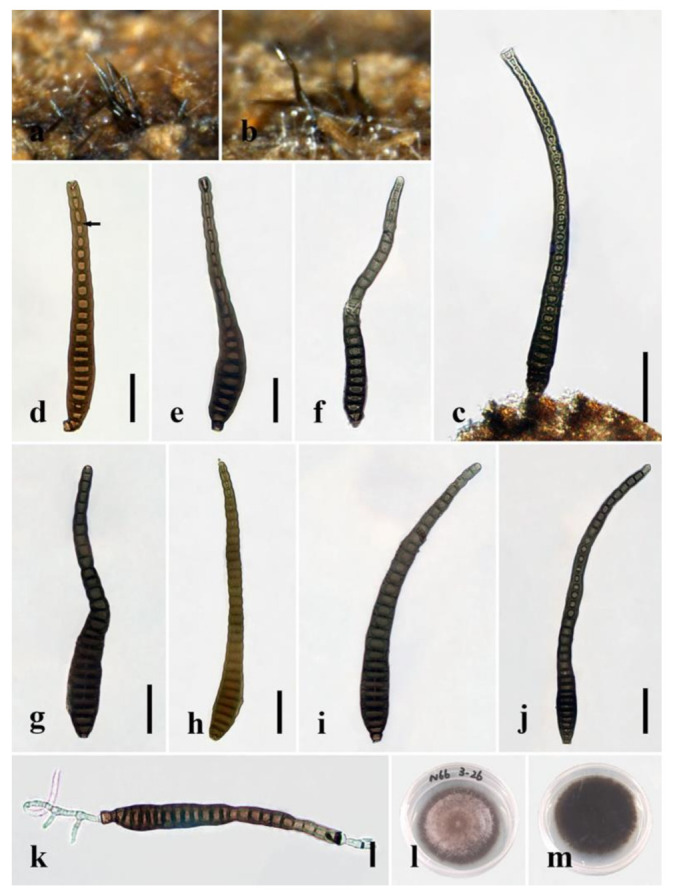
*Distoseptispora crassispora* (HKAS 122181, holotype). (**a**,**b**) Colonies on natural substrate. (**c**) Conidiophores with conidium. (**d**–**j**) Conidia. The arrow in **d** points to a gap in the middle of the septum, which indicates the distosepta. (**k**) Germinated conidium. (**l**) Colony on PDA (from front). (**m**) Colony on PDA (from reverse). Scale bars: (**c**) 40 μm; (**d**–**f**) 20 μm; (**g**–**j**) 30 μm; (**k**) 2 μm.

**Figure 8 jof-08-01063-f008:**
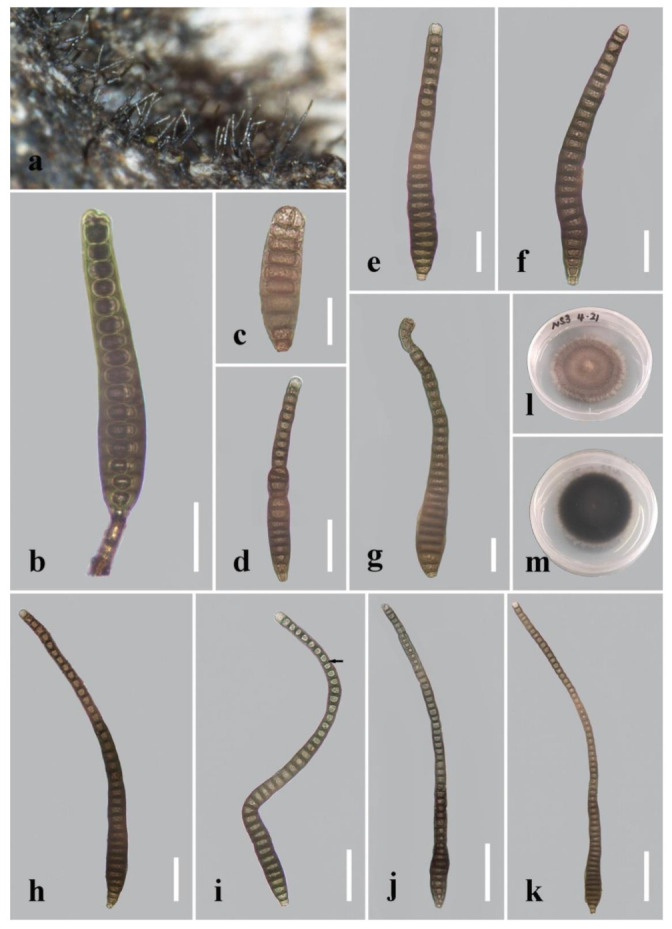
*Distoseptispora curvularia* (HKAS 122180, holotype). (**a**) Colonies on natural substrate. (**b**) Conidiophores with conidium. (**c**–**k**) Conidia. The arrow in **i** points to a gap in the middle of the septum, which indicates the distosepta. (**l**) Colony on PDA (from front). (**m**) Colony on PDA (from reverse). Scale bars: (**b**–**g**) 20 μm; (**h**–**i**) 30 μm; (**j**–**k**) 50 μm.

**Figure 9 jof-08-01063-f009:**
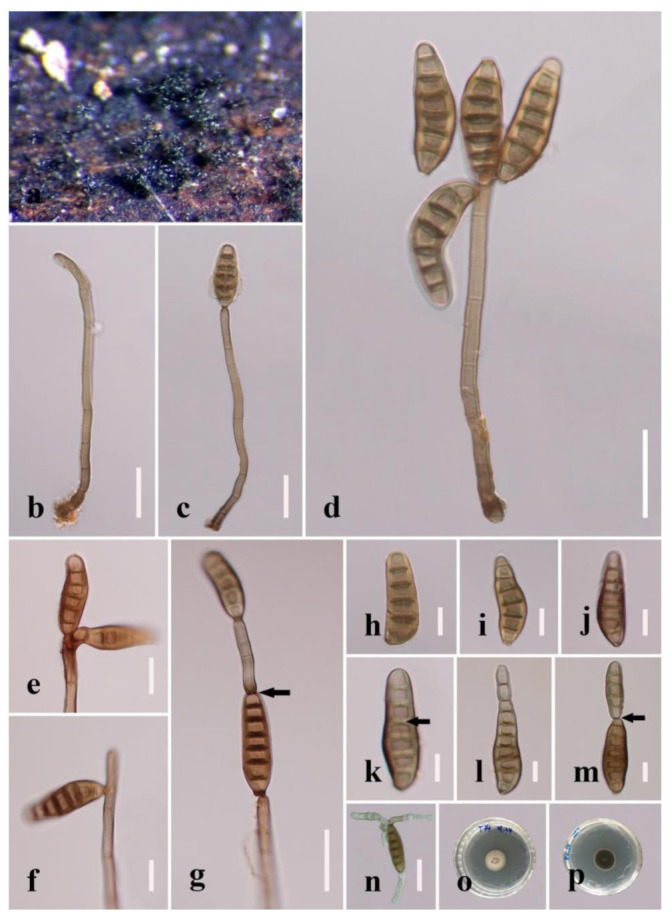
*Distoseptispora dehongensis* (from MFLU 19–0544). (**a**) Colonies on natural substrate. (**b**) Conidiophore. (**c**,**d**) Conidiophores with conidia. (**e**–**g**) Conidiogenous cells bearing conidia. Note the conidium with percurrent proliferation that is forming another conidium in **g**. (**h**–**m**) Conidia. Arrow in **k** points a gap in the middle of septum, which indicateds the distoseptate, note the percurrently proliferating conidium in **m**; (**n**) Germinated conidium. (**o**) Colony on PDA (from front). (**p**) Colony on PDA (from reverse). Scale bars: (**b**–**c**,**n**) 20 μm; (**d**,**g**) 30 μm; (**e**,**f**,**h**–**m**) 10 μm.

**Figure 10 jof-08-01063-f010:**
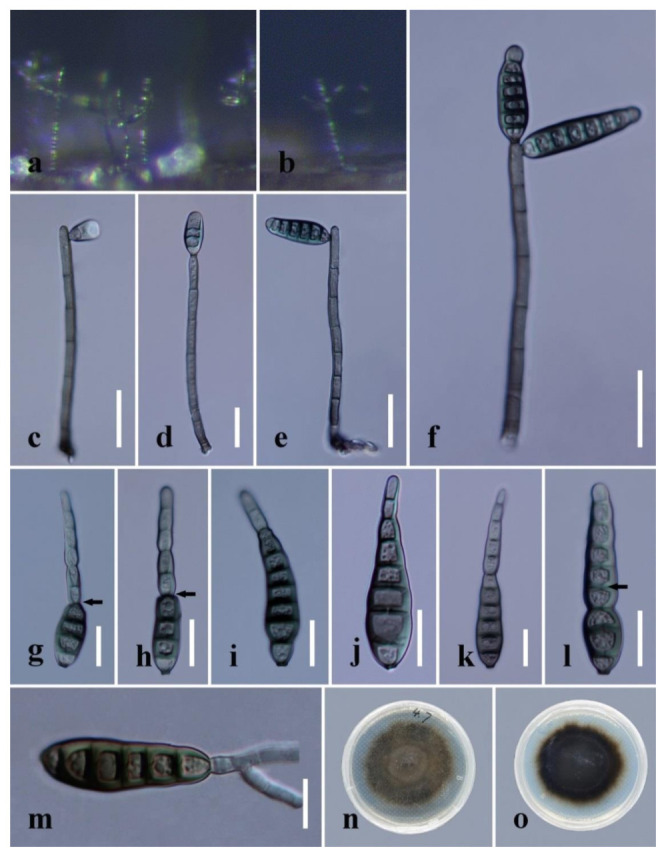
*Distoseptispora dehongensis* (from MFLU 17–1671). (**a**,**b**) Colonies on natural substrate. (**c**–**f**) Conidiophores with conidia. (**g**–**l**) Conidia. The arrows in **g** and **h** indicate conidial percurrent proliferating points. The arrow in **l** points to a gap in the middle of the septum, which indicates the distosepta. (**m**) Germinated conidium. (**n**) Colony on PDA (from front). (**o**) Colony on PDA (from reverse). Scale bars: (**c**–**f**) 20 μm; (**g**–**l**) 15 μm; (**m**) 10 μm.

**Figure 11 jof-08-01063-f011:**
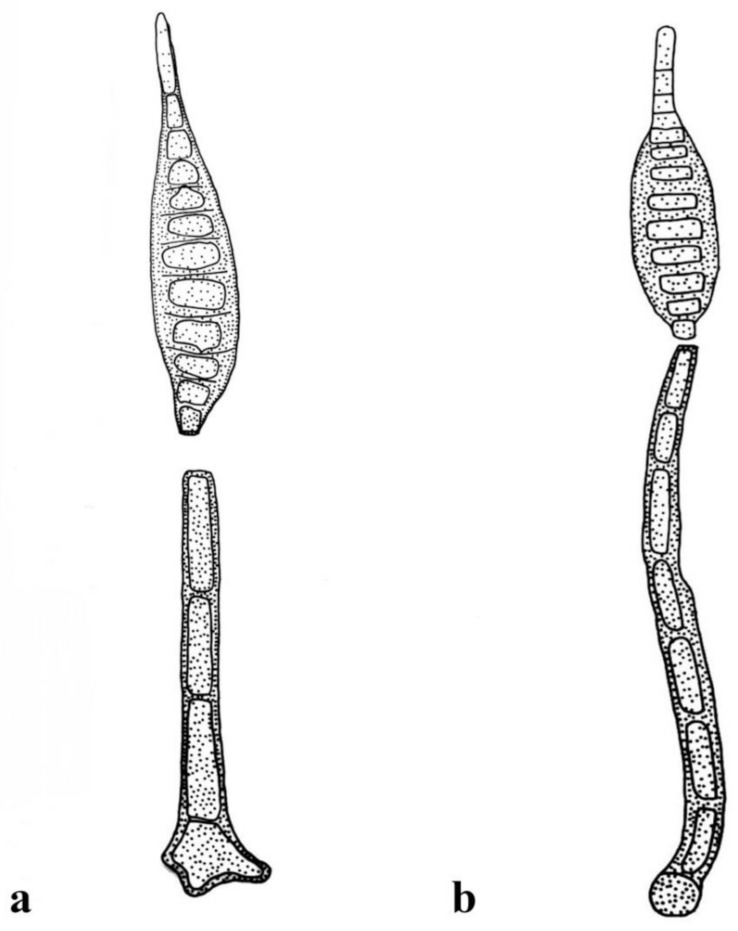
*Distoseptispora leonensis* (redraw according to Wu and Zhuang [14]). (**a**,**b**) Conidiophores with conidia.

**Figure 12 jof-08-01063-f012:**
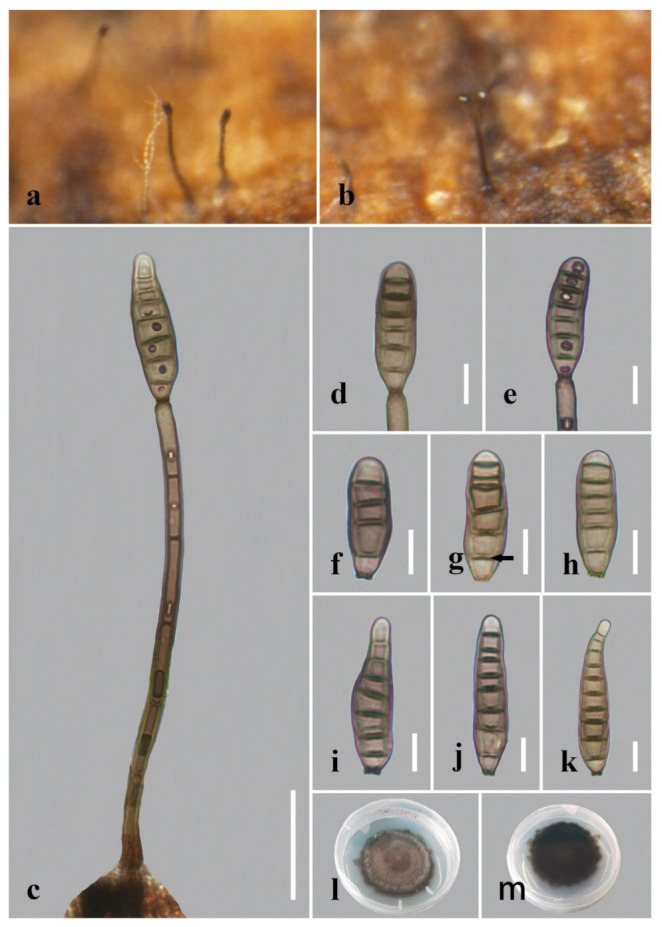
*Distoseptispora nonrostrata* (HKAS 122185, holotype). (**a**,**b**) Colonies on natural substrate. (**c**) Conidiophore with conidium. (**d**,**e**) Conidiogenous cells bearing conidia. (**f**–**k**) Conidia. The arrow in **g** points to a gap in the middle of the septum, which indicates the distosepta. (**l**) Colony on PDA (from front). (**m**) Colony on PDA (from reverse). Scale bars: (**c**) 30 μm; (**d**–**k**) 10 μm.

**Figure 13 jof-08-01063-f013:**
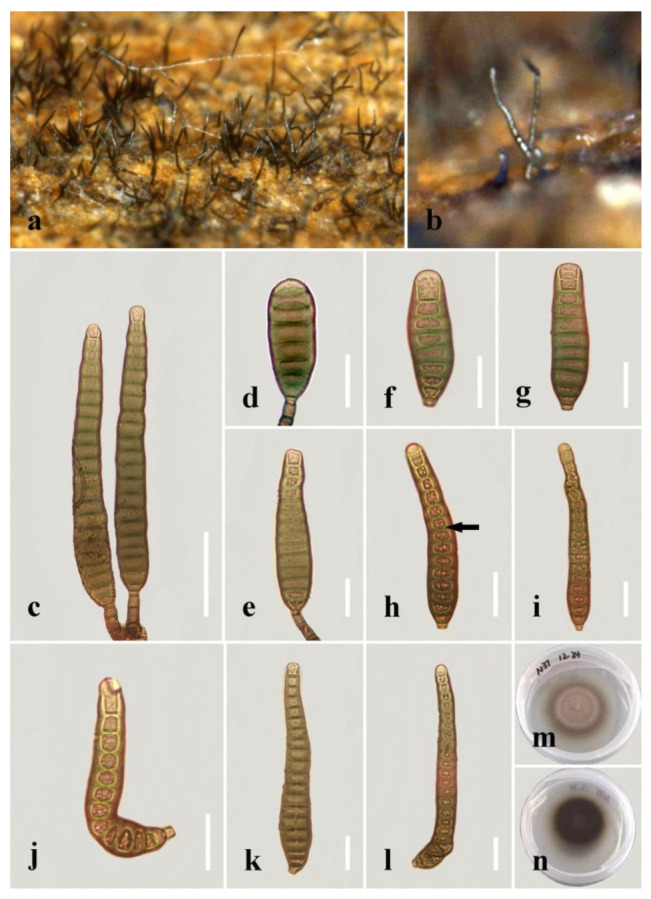
*Distoseptispora pachyconidia* (HKAS 122179, holotype). (**a**,**b**) Colonies on natural substrate. (**c**) Conidiophores with conidia. (**d**,**e**) Conidiogenous cells with conidia. (**f**–**l**) Conidia. The arrow in **h** points to a gap in the middle of the septum, which indicates the distosepta. (**m**) Colony on PDA (from front). (**n**) Colony on PDA (from reverse). Scale bars: (**c**) 40 μm; (**d**–**l**) 20 μm.

**Figure 14 jof-08-01063-f014:**
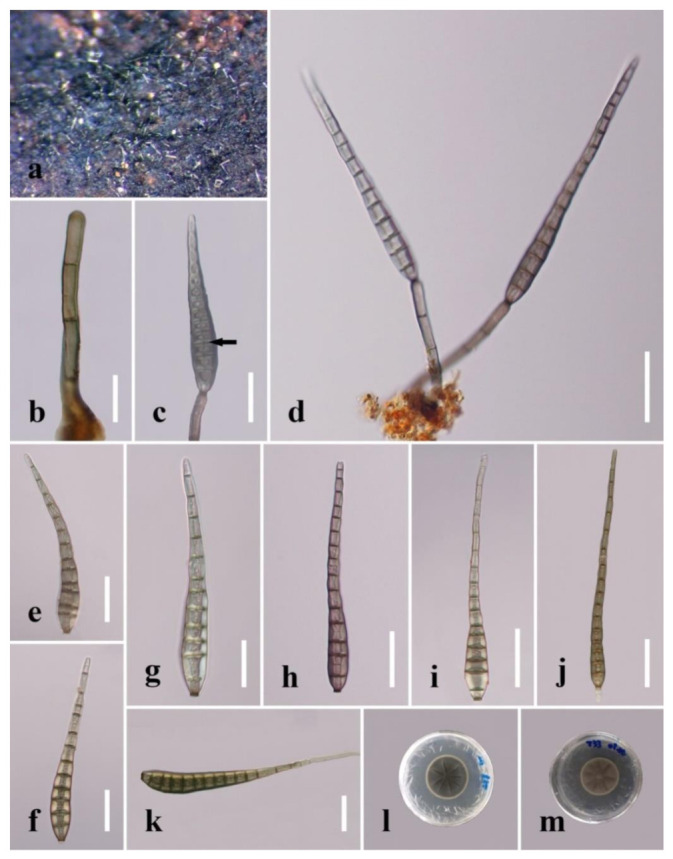
*Distoseptispora rayongensis*. (**a**) Colonies on natural substrate. (**b**) Conidiophore. (**c**) Conidiogenous cell with conidium. The arrow in **c** points to a gap in the middle of the septum, which indicates the distosepta. (**d**) Conidiophores with conidia. (**e**–**j**) Conidia. (**k**) Germinated conidium. (**l**) Colony on PDA (from front). (**m**) Colony on PDA (from reverse). Scale bars: (**b**) 20 μm; (**c**,**e**–**k**) 30 μm; (**d**) 40 μm.

**Figure 15 jof-08-01063-f015:**
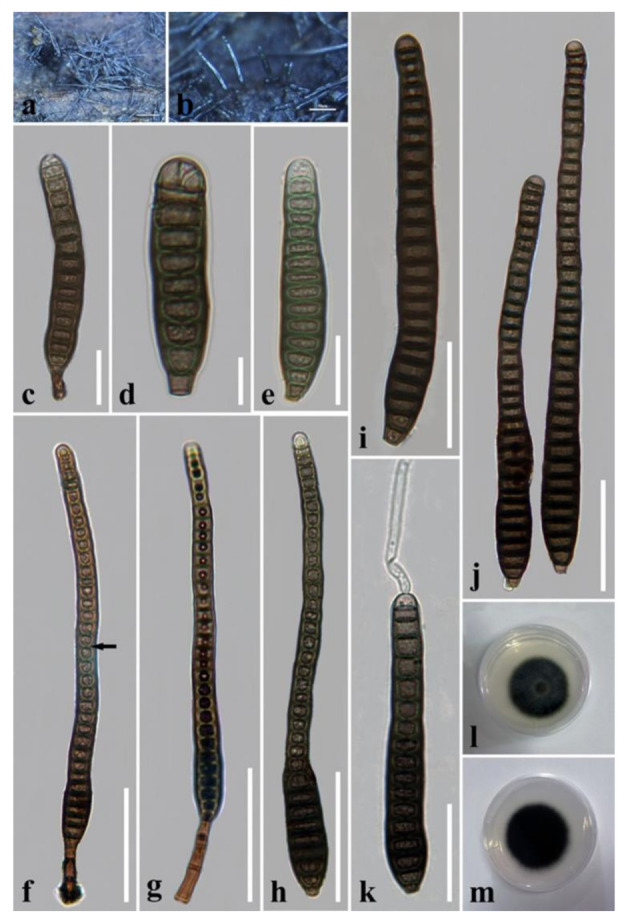
*Distoseptispora tectonae*. (**a**,**b**) Colonies on natural substrate. (**c**,**f**,**g**) Conidiophores with conidia. The arrow in **f** points to a gap in the middle of the septum, which indicates the distosepta. (**d**,**e**,**h**–**j**) Conidia. (**k**) Germinated conidium. (**l**) Colony on PDA (from front). (**m**) Colony on PDA (from reverse). Scale bars: (**c**,**e**) 20 μm; (**d**) 10 μm; (**f**–**h**) 50 μm; (**i**,**k**) 30 μm; (**j**) 40 μm.

**Table 1 jof-08-01063-t001:** The primers for the LSU, ITS, *TEF1*-α and *RPB2* genes and the PCR thermal cycle programs used in this study.

Gene	PCR Primers	PCR Amplification Reaction System	PCR Conditions
LSU, ITS	LR5/LROR [28], ITS5/ITS4 [29]	Total: 25 μL 12.5 μL Master Mix 9.5 μL ddH2O 1 μL of each primer (10 μM) 1 μL DNA template	1. 94 °C, 3 min 2. 94 °C, 45 s 3. 56 °C, 50 s 4. 72 °C, 1 min 5. Repeat steps 2–4 for 35 cycles 6. 72 °C, 10 min 7. Keep at 4 °C
*TEF1*-α	983F/2218R [30]	Total: 50 μL 45 μL T3 Mix 2 μL of each primer (10 μM) 1 μL DNA template	1. 98 °C, 2 min 2. 98 °C, 10 s 3. 56 °C, 10 s 4. 72 °C, 20 s 5. Repeat steps 2–4 for 35 cycles 6. 72 °C, 5 min 7. Keep at 4 °C
*RPB2*	5F/7cR [31]	Total: 50 μL 45 μL T3 Mix 2 μL of each primer (10 μM) 1 μL DNA template	1. 98 °C, 2 min 2. 98 °C, 10 s 3. 51 °C, 10 s 4. 72 °C, 20 s 5. Repeat steps 2–4 for 35 cycles 6. 72 °C, 5 min 7. Keep at 4 °C

**Table 2 jof-08-01063-t002:** Taxa used in the phylogenetic analyses and their corresponding GenBank accession numbers.

Species	Voucher/Culture	GenBank Accession Numbers			
		LSU	ITS	*TEF**1*-α	*RPB2*
*Acrodictys bambusicola*	CGMCC 3.18641	KX033564	KU999973	–	–
*Acrodictys elaeidicola*	CGMCC 3.18642	KX033569	KU999978	–	–
*Aquapteridospora aquatica*	MFLUCC 17–2371^T^	MW287767	MW286493	–	–
*Aquapteridospora fusiformis*	MFLU 18–1601^T^	MK849798	MK828652	MN194056	–
*Aquapteridospora lignicola*	MFLUCC 15–0377^T^	KU221018	MZ868774	MZ892980	MZ892986
*Bullimyces aurisporus*	AF316–1b^T^	JF775590	–	–	–
*Bullimyces communis*	AF281–5	JF775587	–	–	–
*Cancellidium applanatum*	CBS 337.76	MH872755	NR159777	_	_
*Cancellidium cinereum*	MFLUCC 18–0424^T^	MT370363	MT370353	MT370488	MT370486
*Distoseptispora adscendens*	HKUCC 10820	DQ408561	–	–	DQ435092
*Distoseptispora amniculi*	MFLUCC17–2129^T^	MZ868761	MZ868770	–	MZ892982
*Distoseptispora appendiculata*	MFLUCC 18–0259^T^	MN163023	MN163009	MN174866	–
** *Distoseptispora aqualignicola* **	**KU** **N** **CC 21–10729^T^**	**ON40845**	**OK341186**	**OP413480**	**OP413474**
** *Distoseptispora aquamyces* **	**KUNCC 21–10732^T^**	**OK341199**	**OK341187**	**OP413482**	**OP413476**
*Distoseptispora aquatica*	MFLUCC 15–0374^T^	KU376268	MF077552	–	–
*Distoseptispora aquatica*	MFLUCC 16–0904	MK849794	MK828649	MN194053	–
*Distoseptispora aquatica*	MFLUCC 18–0646	MK849793	MK828648	MN194052	–
*Distoseptispora aquatica*	S-965	MK849792	MK828647	MN194051	MN124537
*Distoseptispora atroviridis*	GZCC 20–0511^T^	MZ868763	MZ868772	MZ892978	MZ892984
*Distoseptispora atroviridis*	GZCC 19–0531	MZ227223	MW133915	–	–
*Distoseptispora bambusae*	MFLUCC 20–0091^T^	MT232718	MT232713	MT232880	MT232881
*Distoseptispora bambusae*	MFLUCC 14–0583	MT232717	MT232712	–	MT232882
** *Distoseptispora bambusae* **	**KU** **N** **CC 21–10732**	**OK341200**	**OK341188**	**OP4134** **92**	**OP413487**
*Distoseptispora bangkokensis*	MFLUCC 18–0262^T^	MZ518206	MZ518205	–	**–**
*Distoseptispora cangshanensis*	MFLUCC 16–0970 ^T^	MG979761	MG979754	MG988419	–
*Distoseptispora caricis*	CPC 36498^T^	MN567632	MN562124	–	MN556805
*Distoseptispora caricis*	CPC 36442^T^	–	MN562125	–	MN556806
*Distoseptispora chinense*	GZCC 21–0665 ^T^	MZ474867	MZ474871	MZ501609	–
*Distoseptispora clematidis*	MFLUCC 17–2145^T^	MT214617	MT310661	–	MT394721
** *Distoseptispora clematidis* **	**KUMCC 21–10727**	**OK341197**	**OK341184**	**OP413488**	**OP413483**
** *Distoseptispora crassispora* **	**KUMCC 21–10726^T^**	**OK341196**	**OK310698**	**OP413479**	**OP413473**
** *Distoseptispora curvularia* **	**KUMCC 21–10725^T^**	**OK341195**	**OK310697**	**OP413478**	**OP413472**
*Distoseptispora dehongensis*	KUMCC 18–0090 ^T^	MK079662	MK085061	MK087659	–
** *Distoseptispora dehongensis* **	**MFLUCC 19–0** **335**	**OK341201**	**OK341189**	**OP413491**	**OP413486**
** *Distoseptispora dehongensis* **	**MFLUCC 17–2326**	**OK341193**	**OK341183**	**OP413493**	–
*Distoseptispora effusa*	GZCC 19–0532^T^	MZ227224	MW133916	MZ206156	–
*Distoseptispora euseptata*	MFLUCC 20–0154^T^	MW081544	MW081539	–	MW151860
*Distoseptispora euseptata*	DLUCC S2024	MW081545	MW081540	MW084994	MW084996
*Distoseptispora fasciculata*	KUMCC 19–0081^T^	MW287775	MW286501	MW396656	–
*Distoseptispora fluminicola*	MFLUCC 15–0417^T^	KU376270	NR154041	–	–
*Distoseptispora fusiformis*	GZCC 20–0512^T^	MZ868764	MZ868773	MZ892979	MZ892985
*Distoseptispora guizhouensis*	GZCC 21–0666^T^	MZ474869	MZ474868	MZ501610	MZ501611
*Distoseptispora guttulata*	MFLUCC 16–0183^T^	MF077554	MF077543	MF135651	–
*Distoseptispora hyalina*	MFLUCC 17–2128 ^T^	MZ868760	MZ868769	MZ892976	MZ892981
*Distoseptispora hydei*	MFLUCC 20–0115^T^	MT742830	MT734661	–	MT767128
*Distoseptispora lancangjiangensis*	DLUCC 1864^T^	MW879522	MW723055	–	–
*Distoseptispora leonensis*	HKUCC 10822	DQ408566	–	–	DQ435089
*Distoseptispora lignicola*	MFLUCC 18–0198^T^	MK849797	MK828651	–	–
*Distoseptispora longispora*	HFJAU 0705^T^	MH555357	MH555359	–	–
*Distoseptispora martinii*	CGMCC 3.18651	KX033566	KU999975	–	–
*Distoseptispora meilingensis*	JAUCC 4727^T^	OK562396	OK562390	OK562408	–
*Distoseptispora meilingensis*	JAUCC 4728^T^	OK562397	OK562391	OK562409	–
*Distoseptispora multiseptata*	MFLUCC 16–1044	MF077555	MF077544	MF135652	MF135644
*Distoseptispora multiseptata*	MFLUCC 15–0609^T^	KX710140	KX710145	MF135659	–
*Distoseptispora neorostrata*	MFLUCC 18–0376^T^	MN163017	MN163008	–	–
** *Distoseptispora nonrostrata* **	**KUNCC 21–10730^T^**	**OK341198**	**OK310699**	**OP413481**	**OP413475**
*Distoseptispora obclavata*	MFLUCC 18–0329^T^	MN163010	MN163012	–	–
*Distoseptispora obpyriformis*	DLUCC 0867	MG979765	MG979757	MG988423	MG988416
*Distoseptispora obpyriformis*	MFLUCC 17–1694^T^	MG979764	–	MG988422	MG988415
** *Distoseptispora pachyconidia* **	**KUMCC 21–10724^T^**	**OK341194**	**OK310696**	**OP413477**	**OP413471**
*Distoseptispora palmarum*	MFLUCC 18–1446^T^	MK079663	MK085062	MK087660	MK087670
*Distoseptispora phangngaensis*	MFLUCC 16–0857^T^	MF077556	MF077545	MF135653	–
*Distoseptispora rayongensis*	MFLUCC 18–0417	MH457138	MH457173	MH463254	MH463256
*Distoseptispora rayongensis*	MFLUCC 18–0415^T^	MH457137	MH457172	MH463253	MH463255
** *Distoseptispora rayongensis* **	**MFLU 19–0543**	**MN163010**	**MN513037**	**OP413490**	**OP413485**
*Distoseptispora rostrata*	DLUCC 0885	MG979767	MG979759	MG988425	–
*Distoseptispora rostrata*	MFLUCC 16–0969^T^	MG979766	MG979758	MG988424	MG988417
*Distoseptispora saprophytica*	MFLUCC 18–1238^T^	MW287780	MW286506	MW396651	MW504069
*Distoseptispora songkhlaensis*	MFLUCC 18–1234^T^	MW287755	MW286482	MW396642	–
*D* *istoseptispora suoluoensis*	MFLUCC 17–1305	MF077558	MF077547	–	MZ945510
*D* *istoseptispora suoluoensis*	MFLUCC 17–0224^T^	MF077557	MF077546	MF135654	–
*Distoseptispora tectonae*	MFLUCC 12–0291^T^	KX751713	KX751711	KX751710	KX751708
*Distoseptispora tectonae*	MFLUCC 15–0981	MW287763	MW286489	MW396641	–
*Distoseptispora tectonae*	MFLUCC 16–0946	MG979768	MG979760	MG988426	MG988418
** *Distoseptispora tectonae* **	**KUNCC 21–10728**	**OK348852**	**OK341185**	**OP413489**	**OP413484**
*Distoseptispora tectonigena*	MFLUCC 12–0292^T^	KX751714	KX751712	–	KX751709
*Distoseptispora thailandica*	MFLUCC 16–0270^T^	MH260292	MH275060	MH412767	–
*Distoseptispora thysanolaenae*	KUMCC 18–0182^T^	MK064091	MK045851	MK086031	–
*Distoseptispora thysanolaenae*	HKAS 112710	MW879524	MW723057	MW729783	–
*Distoseptispora verrucosa*	GZCC 20–0434^T^	MZ868762	MZ868771	MZ892977	MZ892983
*Distoseptispora xishuangbannaensis*	KUMCC 17–0290^T^	MH260293	MH275061	MH412768	MH412754
*Distoseptispora yongxiuensis*	JAUCC 4725^T^	OK562394	OK562388	OK562406	–
*Distoseptispora yongxiuensis*	JAUCC 4726^T^	OK562395	OK562389	OK562407	–
*Distoseptispora yunjushanensis*	JAUCC 4724^T^	OK562398	OK562392	OK562410	–
*Distoseptispora yunjushanensis*	JAUCC 4723^T^	OK562399	OK562393	OK562411	–
*Distoseptispora yunnanensis*	MFLUCC 20–0153^T^	MW081546	MW081541	MW081541	MW151861
*Fluminicola saprophytica*	MFLUCC 15–0976^T^	MF374367	MF374358	MF370956	MF370954
*Myrmecridium banksiae*	CPC 19852^T^	JX069855	JX069871	–	–
*Myrmecridium schulzeri*	CBS 100.54	EU041826	EU041769	–	–
*Papulosa amerospora*	AFTOL-ID 748	DQ470950	–	DQ471069	DQ470901
*Pleurophragmium bambusinum*	MFLUCC 12–0850	KU863149	KU940161	KU940213	–
*Pseudostanjehughesia aquitropica*	MFLUCC 16–0569^T^	MF077559	MF077548	MF135655	–
*Pseudostanjehughesia lignicola*	MFLUCC 15–0352^T^	MK849787	MK828643	MN194047	MN124534
*Wongia griffinii*	DAR 80512^T^	KU850471	KU850473	–	–

Notes: The ex-type cultures are indicated using “^T^” after strain numbers; newly generated sequences are indicated in bold. “–” stands for no sequence data in GenBank.

## Data Availability

The datasets generated for this study can be found in the NCBI database.

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
