# Peer review of "Polyphasic Identification of Distoseptispora with Six New Species from Fresh Water"

_jof, 2022, doi:10.3390/jof8101063_

Round 1

Reviewer 1 Report

This paper is well written however, below, primary concerns need to be addressed before accepting this manuscript

1) The material is corrected from Thailand, and the author listed some of the material collected from Thailand and deposited in the Mae Fah Luang University. I am wondering; there is no single author from Thailand. How did you collect these samples? And what is the agreement to bring them to China as China has too much-restricted material to enter China? What is the MTA

2) You did not generate a single sequence of RPB2. However, the tree is based on that region as well. The tree needs to be redone, removing the RPB2 region. 

3) Morphologically, Distoseptispora aqualignicola and Distoseptispora aquamyces are indistinguishable, and there is no phylogenetic support to say they are different.  More evidence is essential; otherwise, this need to keep as a single species.

4) more evidence need to discuss in the notes to provide crassispora and pachyconidia

Author Response

Response to Reviewer 1

Question 1) The material is corrected from Thailand, and the author listed some of the material collected from Thailand and deposited in the Mae Fah Luang University. I am wondering; there is no single author from Thailand. How did you collect these samples? And what is the agreement to bring them to China as China has too much-restricted material to enter China? What is the MTA

Response: Thank you for your question. In the text, three specimens of two known species were collected from Thailand, viz. D. dehongensis: MFLU 19–0544 and MFLU 17–1671, D. rayongensis: MFLU 19–0543. They were collected by the fourth author Hao Yang in 2018. Yang went to MFLU, Thailand as a visitor student in 2018 which supported by his scholarship from Kunming University of Science & Technology. I will add his scholarship in acknowledgment. He did not bring those specimens back to China but sequences.

Question 2) You did not generate a single sequence of RPB2. However, the tree is based on that region as well. The tree needs to be redone, removing the RPB2 region.

Response: Thank you for your question. We have TEF1 and RPB2 sequences of all species, but we have not got GenBank code for them. I will attach the sequences.

Question 3) Morphologically, Distoseptispora aqualignicola and Distoseptispora aquamyces are indistinguishable, and there is no phylogenetic support to say they are different.  More evidence is essential; otherwise, this need to keep as a single species.

Response: Thank you for your question. Morphologically, D. aqualignicola possesses smooth-walled conidia, which is distinct with verrucose conidia in D. aquamyces. In the phylogenetic tree, D. aquamyces is close to D. suoluoensis rather than D. aqualignicola.  Therefore, we introduced D. aqualignicola and D. aquamyces as different species.

Question 4) more evidence need to discuss in the notes to provide crassispora and pachyconidia

Response: Thank you for your suggestion.

The notes of D. crassispora were revised as “Multi-gene phylogenetic analyses show that D. crassispora is a distinct species in Distoseptispora and clusters with D. chinense and D. tectonigena (97% MLBS/1.00 BIPP, Figure 1). Distoseptispora crassispora is morphologically similar to these two species in having straight to slightly curved, septate conidiophores and obclavate, distoseptate, rostrate, straight or slightly curved conidia. However, D. crassispora possesses shorter conidiophores (up to 27 μm vs. up to 110 μm ) and wider conidia (13–24 μm vs. 10–13 μm) than D. tectonigena. Additionally, the conidia of D. tectonigena are sometimes percurrently proliferating 5–10 times at apex, which was not observed in D. crassispora [15]. Distoseptispora crassispora can be distinguished from D. chinense in molecular data. Comparisons of nucleotide between D. crassispora and D. chinense reveal 12 (2.3%, including 4 gaps) and 23 (2.6%) nucleotide diferences in ITS and TEF1–α, respecvtively,. which follows the generally accepted norm that more than 1.5% of nucleotide differences in the ITS region is likely to be a new species [40]. We, therefore, introduce D. crassispora as a new species in Distoseptispora.”

The notes of D. pachyconidia were revised as “Distoseptispora pachyconidia clusters as an independent branch in Distoseptispora based on concatenated LSU-ITS-TEF1–α-RPB2 phylogeny. It phylogenetically closes to D. crassispora, D. chinense and D. tectonigena. Compared to D. tectonigena, D. pachyconidia has shorter conidiophores (11–27 μm vs. up to 110 μm) and shorter conidia (42–136 μm vs. 148–225(–360) μm), as well as less conidial septa (8–21 vs. 20–46) [3]. Besides, the conidia of D. tectonigena were sometimes percurrently proliferating 5–10 times at apex, which was not observed in D. pachyconidia. The conidia in D. pachyconidia are lighte-colored and shorter than those in D. crassispora and D. chinense (42–136 μm vs. 95–197μm vs.  81–283) [42]. More importantly, D. pachyconidia phylogeneticall distinguished from D. crassispora, D. chinense and D. tectonigena.”

Reviewer 2 Report

Dear Authors

The manuscript entitled “Polyphasic Identification of Distoseptispora with Six New Species from Freshwater” is interesting to me. The manuscript is well-worth publishing. I think the study is good and the data presented in the manuscript is sufficient but I have noticed that the manuscript needs some minor improvement that I pointed out in the text. The content of the paper could be published in JoF. Also, the uniformity of referencing format is not respected in some parts of the text.

Best regards

Author Response

Response to Reviewer 2

  1. should be Italic!!
  2. Italic

Response: Thank you for your suggestion. We have revised.

  1. Please specify final concentration of these!!!

Response: Thank you for your suggestion. We have added the concentration of primer as “10 μM”.

DNA was extracted from fresh mycelium according to the manufacturer’s instructions of the Plant genomic DNA extraction kit (generic) (TreliefJM Kunming, P.R. China). We did not analyze the concentration of DNA.

  1. Do not use abbreviations at the beginning of sentences

Response: Thank you for your suggestion. We have revised.

  1. Strong

Response: Thank you for your suggestion. We have revised.

  1. Please provide them!

Response: Thank you for your suggestion. We did not apply these codes when submitted the manuscript because we afraid the names or description need to be revised. We will apply them once they are confirmed.

  1. invalidly!!?

Response: Thank you for your question. We have revised as “validly”.

  1. In the fig. 1 the font size is small and the quality is low, please improve it!

Response: Thank you for your suggestion. The small font of the characters is because there are too many characters we want to show. They are hard to arrange if change them bigger. The readers can zoom in to see them clearly. We inserted a picture with good quality, but it was after saved. We cut the figure into two pictures and insert a high quality picture again. We also attach it in PDF file.

  1. Please, Separate this paragraph from the Fig. 1 caption

Response: Thank you for your suggestion. It was separate from the Fig. 1 caption in our submitted manuscript. We separate it again.

  1. adscendens

Response: Thank you for your suggestion. We have revised.

  1. Please, Separate this paragraph from the Fig. 2 caption

Same Response as Question 9.

  1. Please provide them!

Response: Thank you for your suggestion. We have added.

  1. Please, Separate this paragraph from the Fig. 3 caption

Same Response as Question 9.

  1. Please provide them!

Response: Thank you for your suggestion. We have added.

  1. Please, Separate this paragraph from the Fig. 4 caption

Same Response as Question 9

  1. This sentence is a repetition of the above sentence. Please avoid repeating sentences.

Response: Thank you for your suggestion. We have deleted.

  1. Please, Separate this paragraph from the Fig. 5 caption
  2. Please, Separate this paragraph from the Fig. 6 caption

Same Response as Question 9

  1. Strong
  2. strong
  3. Italic
  4. leonensis
  5. leonensis
  6. Strong
  7. strong

Response: Thank you for your suggestion. We have revised.
